# Language-Guided Traffic Simulation via Scene-Level Diffusion

**Ziyuan Zhong**[1], **Davis Rempe**[2,3], **Yuxiao Chen**[2], **Boris Ivanovic**[2],
**Yulong Cao**[2], **Danfei Xu**[2,4], **Marco Pavone**[2,3], **Baishakhi Ray**[1]

[1]Columbia University,   [2]NVIDIA Research,   [3]Stanford University,   [4]Georgia Tech

**Abstract:** Realistic and controllable traffic simulation is a core capability that
is necessary to accelerate autonomous vehicle (AV) development. However, cur-
rent approaches for controlling learning-based traffic models require significant
domain expertise and are difficult for practitioners to use. To remedy this, we
present CTG++, a scene-level conditional diffusion model that can be guided by
language instructions. Developing this requires tackling two challenges: the need
for a realistic and controllable traffic model backbone, and an effective method
to interface with a traffic model using language. To address these challenges, we
first propose a scene-level diffusion model equipped with a spatio-temporal trans-
former backbone, which generates realistic and controllable traffic. We then har-
ness a large language model (LLM) to convert a user's query into a loss function,
guiding the diffusion model towards query-compliant generation. Through com-
prehensive evaluation, we demonstrate the effectiveness of our proposed method
in generating realistic, query-compliant traffic simulations.

**Keywords:** Traffic Simulation, Multi-Agent Diffusion, Large Language Model

## 1   Introduction

Given the high costs and risks of large-scale real-world autonomous vehicle (AV) testing [1, 2],
AV developers increasingly rely on simulations for developing robust systems [3]. For maximum
efficacy, simulators must offer *realistic* and *controllable* traffic behaviors, complemented by a *user-
friendly interface*. The realism of traffic patterns ensures that development and testing conducted
in simulation environments can be transferred to real-world scenarios. Controllability permits the
generation of relevant traffic scenarios to scrutinize specific AV behaviors. For example, controlling
a vehicle to collide with the AV to check how it reacts in dangerous situations. A user-friendly
interface simplifies how desired behaviors can be specified. However, generating realistic [4, 5] and
controllable [6] traffic poses considerable challenges, and the exploration of user-friendly interfaces
in traffic generation has been limited. This work strives to develop an expressive scene-centric traffic
model that can be controlled through a user-friendly text-based interface. Such an interface has the
potential to connect simulation to previously unusable text-based data, such as governmental and
insurance collision reports. It also facilitates new simulation capabilities, such as reconstructing
real-world collision scenarios [7].

Building a traffic simulation model with a language interface presents two challenges. First, the
traffic model must generate realistic trajectories at both agent and scene levels, and provide con-
trollability over its generated trajectories. Current simulators [8, 9, 10], whether replaying logs
or using heuristic controllers for agent behavior, lack *realism* and expressiveness. Data-driven ap-
proaches [4, 5] merely reflect training data distribution, lacking *control* over generated traffic. Re-
cently, CTG [6] applies a diffusion model, which has demonstrated promising results across various
conditional generation tasks [11, 12, 13, 14, 15], to traffic generation. CTG shows that diffusion is
well-suited for controllable traffic simulation through *guidance*, which allows test-time adaptability
to user controls. However, CTG models agents independently, leading to unrealistic interactions.

7th Conference on Robot Learning (CoRL 2023), Atlanta, USA.

For example, two vehicles modeled separately might collide if the leading vehicle slows without the following vehicle responding. The second challenge is grounding language in a powerful traffic simulation backbone, since language conveys more abstract patterns (e.g., "traffic jam" or "following") while traffic models operate on low-level trajectories. To address similar issues, recent research on Large Language Models (LLMs) for robotic behaviors [16, 17] designs a suite of high-level functions (e.g., "pick up" and "use item") that an LLM can employ to control the robot in order to achieve a user-specified task (e.g., "make an omelette"). Essentially, these high-level functions bridge textual instructions and robotic behaviors. Unfortunately, this approach cannot be directly used for realistic traffic simulation. It is infeasible for an LLM to only use a few high-level functions (e.g., "go to location") to generate the entire low-level human-like trajectories.

In this work, we propose a model called CTG++ (see Figure 1) to overcome the aforementioned challenges. To achieve a realistic and adaptable traffic model, our approach harnesses the strengths of diffusion and significantly enhances its capability to cater to multi-agent scenarios. This is achieved through a newly proposed scene-level conditional diffusion model, underpinned by a spatial-temporal transformer architecture. This architecture applies alternating temporal and spatial at-

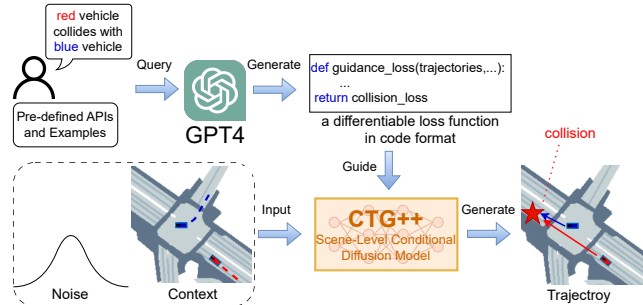

Figure 1: Overview of CTG++. A user query, predefined APIs, and examples are passed to GPT4, which generates a differentiable loss to guide CTG++ for query-compliant trajectories.

tention modules, effectively capturing the dynamics of multi-agent interactions over time. To create a natural language interface with the traffic model, we leverage the proven capacity of LLMs to generate code from natural language [18]. Instead of a direct translation from text to traffic, we introduce an intermediate representation: a differentiable loss function, which encodes a user's intention from the language command. The loss function guides the diffusion model to generate command-compliant trajectories. With this two-step translation, the LLM and diffusion model efficiently bridge the gap between user intent and traffic simulation.

We evaluate CTG++ on the nuScenes dataset [19], showing its ability to follow user-specified language commands and generate realistic trajectories. In summary, our contributions are: **(1)** a scene-level conditional diffusion model, leveraging a spatial-temporal transformer backbone, designed for the generation of realistic and controllable traffic, **(2)** a language interface adept at generating trajectories that align with user-defined rules in language, and **(3)** extensive evaluation comparing CTG++ to state-of-the-art baselines, highlighting its superiority in generating high-quality scenarios.

## 2   Related Work

**Traffic simulation.** Traffic simulation methods can be divided into rule-based and learning-based. Rule-based strategies often feature an interface allowing users to specify the vehicles' routes, where the motion is governed by analytical models like intelligent driver model [20]. Although they deliver user-friendly controllability, their behavioral expressiveness is limited, resulting in trajectories far from human driving patterns. To improve realism, learning-based approaches utilize deep generative models trained on trajectory datasets, aiming to emulate authentic driving behaviors [21, 22, 23, 4, 5]. However, they trade off user-controllability for increased realism, as users cannot customize the properties of generated trajectories. In contrast, our scene-level diffusion model and LLM-based language interface allow the generation of realistic and language command-compliant traffic.

**Diffusion models for conditional generation in robotics.** As diffusion models have shown strong performance and test-time adaptability, they have been recently used for various conditional generation tasks in robotics and traffic. Existing works use trained classifiers [13, 14] or expert-designed

loss functions [6, 24, 25] to guide the denoising process to achieve user-desired properties. For example, CTG [6], the closest work to ours, uses a manually designed loss function based on Signal Temporal Logic (STL) to guide denoising. However, both training a classifier and manually crafting a loss function for each new property require domain expertise. In contrast, our approach allows a user to easily specify desired properties in natural language, which is then converted into a relevant loss function by an LLM. Moreover, most works, including CTG, model each agent independently [26, 13, 14, 6], resulting in unrealistic agent interactions. In contrast, our scene-level diffusion model consider all agents in a scene jointly, resulting in realistic modelling of interactions.

**Large language models for robotics.** Recent breakthroughs in LLMs have motivated a series of works applying LLMs to robotic tasks. One approach is to train a multi-modal LLM that takes in embodied data in addition to text data [27, 28, 29]. Unfortunately, no such text-traffic data is available. Other works directly prompt a pre-trained LLM with a high-level function library along with a user query. This lets the LLM plan a robot's behaviors via the provided functions to achieve the goals in the query [16, 17]. This approach does not directly apply to traffic simulation as existing data-driven approaches cannot be controlled via high-level functions. To tackle this challenge, we leverage a pre-trained LLM to translate a user query into a differentiable loss function in code format and use it to guide a scene-level conditional diffusion model for traffic generation.

## 3 Methodology

After formulating the problem of controllable traffic generation (Section 3.1), we provide the details of our approach, CTG++. The training stage involves training a scene-level conditional diffusion model to capture diverse behaviors from real-world driving data (Section 3.2), utilizing a scene-level spatial-temporal transformer architecture (Section 3.3). During the inference stage, CTG++ generates query-compliant behaviors via the guidance of a user query derived loss (Section 3.4).

### 3.1 Problem Formulation

Similar to CTG [6], we formulate the traffic simulation as an imitation learning problem. For the $M$ vehicles in a scene we want to simulate, let their state at a timestep $t$ be $s_t = [s_t^1 \ ... \ s_t^M]$ where $s_t^i = (x_t^i, y_t^i, v_t^i, \theta_t^i)$ (2D location, speed, and yaw) and the action (*i.e.*, control) be $a_t = [a_t^1 \ ... \ a_t^M]$ where $a_t^i = (\dot{v}_t, \dot{\theta}_t)$ (acceleration and yaw rate). We denote $\mathbf{c} = (I, s_{t-T_{hist}:t})$ to be decision-relevant context, which consists of local semantic maps for all the agents $I = \{I^1, ..., I^M\}$, and their current and $T_{hist}$ previous states $s_{t-T_{hist}:t} = \{s_{t-T_{hist}}, ..., s_t\}$. To obtain state $s_{t+1}$ at time $t+1$, we assume a transition function (e.g., a unicycle dynamics model) $f$ that computes $s_{t+1} = f(s_t, a_t)$ given the previous state $s_t$ and control $a_t$. Our goal is to generate realistic and query-satisfying traffic behavior for the agents given (1) the decision context $\mathbf{c}$ and (2) a function $r : \mathbb{R}^{4T} \times \mathbb{R}^{2T} \rightarrow \mathbb{R}$ derived from a user query to measure rule satisfaction of the state and action trajectories. A model should generate future trajectories for the agents $s_{t+1:t+T}$ over the next $T$ timesteps. Ideally, these trajectories maximize satisfaction $r(a_{t:t+T-1}, s_{t+1:t+T})$ to avoid violating the given rule.

### 3.2 Scene-Level Conditional Diffusion for Traffic Modeling

Diffusion models [30, 31, 13, 14] generate new samples through an iterative denoising process by learning to reverse a diffusion process. As a traffic scene involves multiple traffic participants, a single-agent diffusion model [13, 14, 6] may generate sub-optimal samples when a scene involves significant interactions among multiple agents. To tackle this issue, we propose a scene-level diffusion model that jointly models all traffic participants in a scene. Unlike CTG, which models each agent's future trajectory independently, our model operates on the past and future trajectories of all the agents in a scene jointly (see Figure 2) and thus captures the interactions among agents both spatially and temporally. Starting from Gaussian noise, the diffusion model is applied iteratively to predict a clean, denoised trajectory of states and actions for all agents in a scene.

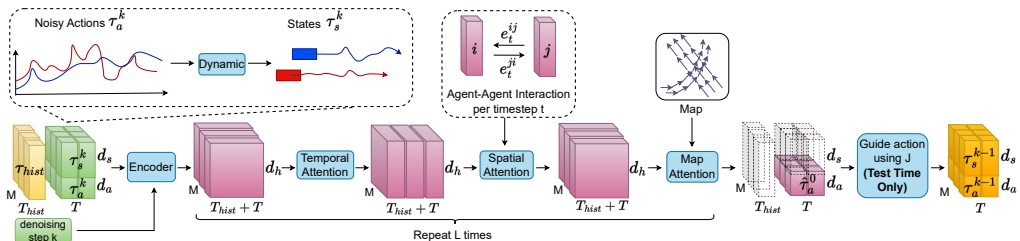

Figure 2: A denoising step using our scene-level spatial-temporal transformer.

**Trajectory Representation.** We denote the future trajectory that the model operates on as:

$$\boldsymbol{\tau} := \begin{bmatrix} \boldsymbol{\tau}_s \\ \boldsymbol{\tau}_a \end{bmatrix}, \quad \boldsymbol{\tau}_a := \begin{bmatrix} \boldsymbol{\tau}_a^1 \\ \vdots \\ \boldsymbol{\tau}_a^M \end{bmatrix}, \quad \boldsymbol{\tau}_s := \begin{bmatrix} \boldsymbol{\tau}_s^1 \\ \vdots \\ \boldsymbol{\tau}_s^M \end{bmatrix}, \quad \boldsymbol{\tau}_a^i := [a_0^i \dots a_{T-1}^i], \quad \boldsymbol{\tau}_s^i := [s_1^i \dots s_T^i].$$

Additionally, we represent the historical trajectory in the context $\mathbf{c}$ as $\boldsymbol{\tau}_{hist}$. In accordance with [6], our model predicts solely the action trajectory $\boldsymbol{\tau}_a$, employing the known dynamics $f$ to deduce the states $\boldsymbol{\tau}_s$ via a rollout from the initial state $s_0$ (which forms part of the context $\mathbf{c}$). To maintain the physical feasibility of the state trajectory throughout the denoising process, we consistently denote $\boldsymbol{\tau}_s$ as a state trajectory resulting from actions: $\boldsymbol{\tau}_s = f(s_0, \boldsymbol{\tau}_a)$.

**Formulation.** Consider $\boldsymbol{\tau}_a^k$ as the action trajectory at the $k$-th diffusion step, with $k = 0$ marking the original clean trajectory. The forward diffusion process that starts from $\boldsymbol{\tau}_a^0$ is defined as:

$$q(\boldsymbol{\tau}_a^{1:K}|\boldsymbol{\tau}_a^0) := \prod_{k=1}^{K} q(\boldsymbol{\tau}_a^k|\boldsymbol{\tau}_a^{k-1}) := \prod_{k=1}^{K} \mathcal{N}(\boldsymbol{\tau}_a^k; \sqrt{1-\beta_k}\boldsymbol{\tau}_a^{k-1}, \beta_k\mathbf{I}). \tag{1}$$

Here, $\beta_i$ for all $i = 1, ..., K$ are a pre-determined variance schedule, controlling the magnitude of noise added at each diffusion step. As the noise incrementally accumulates, the signal transforms into an approximately isotropic Gaussian distribution $\mathcal{N}(\mathbf{0}, \mathbf{I})$. For trajectory generation, our goal is to reverse this diffusion process through a learned conditional denoising model (Figure 2) that is iteratively applied starting from sampled noise. The reverse diffusion process, is as follows:

$$p_\theta(\boldsymbol{\tau}_a^{0:K}|\mathbf{c}) := p(\boldsymbol{\tau}_a^K) \prod_{k=1}^{K} p_\theta(\boldsymbol{\tau}_a^{k-1}|\boldsymbol{\tau}^k, \mathbf{c}) := p(\boldsymbol{\tau}_a^K) \prod_{k=1}^{K} \mathcal{N}(\boldsymbol{\tau}_a^{k-1}; \boldsymbol{\mu}_\theta(\boldsymbol{\tau}^k, k, \mathbf{c}), \boldsymbol{\Sigma}_\theta(\boldsymbol{\tau}^k, k, \mathbf{c})), \tag{2}$$

where $p(\boldsymbol{\tau}_a^K) = \mathcal{N}(\mathbf{0}, \mathbf{I})$ and $\theta$ represents the parameters of the diffusion model. At each step, the model takes in actions $\boldsymbol{\tau}_a^k$ and the resulting states $\boldsymbol{\tau}_s^k = f(s_0, \boldsymbol{\tau}_a^k)$ as input. As per [31, 26], the variance term of the transition is a fixed schedule such that $\boldsymbol{\Sigma}_\theta(\boldsymbol{\tau}^k, k, \mathbf{c}) = \boldsymbol{\Sigma}^k = \sigma_k^2\mathbf{I}$. The training in our approach mirrors that in [6], but with a key difference - trajectories are sampled at the scene level instead of the agent level, as our model simultaneously predicts outcomes for all agents in a scene. Detailed information is available in Appendix A.1.

## 3.3 Model Architecture: Scene-Level Spatial-Temporal Transformer

Unlike previous works which use a U-Net [13, 6] or single-agent transformer [26] to model the denoising process for a single agent, we design an architecture that models multiple agents jointly. Inspired by recent works on transformer-based motion prediction [32, 33, 34, 35, 36], we propose a spatial-temporal transformer architecture to model multiple agents jointly. Unlike most previous work [32, 33], which employs scene-centric coordinates to capture interactions, we adopt agent-centric coordinates. As a traffic scene configuration is combinatorial, it is easy for a scene to end up in an out-of-distribution configuration if it is modeled in scene-centric coordinates, since errors compound over time. In contrast, agent-centric coordinates are invariant to translation and rotation of the scene, and therefore more robust during closed-loop simulation. However, agent-centric coordinates discard relative information among agents, which is important for interactions. To avoid this, we introduce a spatial attention module that enables the exchange of relative information between agents. Inspired by previous work [36], to avoid the combinatoric explosion of attention pairs, we alternate between temporal attention, spatial attention, and map attention module to fully capture the interactions among agents and the map. We next introduce the details of our proposed architecture by showing the data flow of a denoising step (Figure 2; see Figure A1 for more details).

**Input and Temporal Attention.** We first concatenate the ground-truth agent history trajectories with the predicted future trajectories along the temporal dimension and apply a row-wise feed-forward network (rFFN) to project each element (each agent per timestep) from the attribute dimension $d_s + d_a$ to the hidden dimension $d_h$. The denoising step $k$ is injected into the encoded trajectory using a sinusoidal positional encoding function [37]. We next capture the temporal information in the encoded trajectory by feeding it into the temporal attention block, a standard transformer encoder [37] that captures the temporal-wise relation of each agent.

**Spatial Attention.** The encoded trajectory is then fed into a spatial attention block which is a customized transformer decoder block with key and value designed to capture the *relative* geometric relationships among agents. Similar to [35], we extend a regular attention layer to be aware of the relative information $\mathbf{e}_t^{ij}$ between two agents $i$ and $j$ at timestep $t$:

$$\mathbf{e}_t^{ij} = \phi_{\mathrm{r}} \left( \left[ \mathbf{R}_0^{i\top} \left( \Delta x_{0,t}^{ij}, \Delta y_{0,t}^{ij} \right), \cos(\Delta \theta_{0,t}^{ij}), \sin(\Delta \theta_{0,t}^{ij}), v_t^j \cos(\Delta \theta_{0,t}^{ij}) - v_0^i, v_t^j \sin(\Delta \theta_{0,t}^{ij}), d_t^{i,j} \right] \right) \tag{3}$$

where $\phi_{\mathrm{r}}$ is a feed-forward network, $\Delta x_{0,t}^{ij} := x_t^j - x_0^i, \Delta y_{0,t}^{ij} := y_t^j - y_0^i$, and $\Delta \theta_{0,t}^{ij} := \theta_t^j - \theta_0^i$ represent the position and orientation differences from $j$ at timestep $t$ to $i$ at time step $0$ (the current timestep), $d_t^{ij}$ is the relative distance between $i$ and $j$ at timestep $t$, and $\mathbf{R}_0^{i\top}$ is the rotation matrix associated with agent $i$ at timestep $0$. For future timesteps at the training stage and history timesteps, we use the ground-truth relative information. For future timesteps at the inference stage, since we do not have the ground-truth information, we use a constant velocity model (which assumes the agents to keep their current velocity for the future timesteps as in [38]) to estimate the state of all the agents in the future and thus their relative information. The pair-wise relative information is then incorporated into the transformation of the encoded trajectory via keys and values of the decoder layer:

$$\mathbf{q}_t^i = \mathbf{W}^{Q^{\mathrm{global}}} \mathbf{h}_t^i, \quad \mathbf{k}_t^{ij} = \mathbf{W}^{K^{\mathrm{global}}} \left[ \mathbf{h}_t^j, \mathbf{e}_t^{ij} \right], \quad \mathbf{v}_t^{ij} = \mathbf{W}^{V^{\mathrm{global}}} \left[ \mathbf{h}_t^j, \mathbf{e}_t^{ij} \right] \tag{4}$$

where $h_t^i$ and $h_t^j$ are the slices of the encoded trajectories corresponding to the agent $i$ and $j$ at timestep $t$, and $\mathbf{W}^{Q^{\mathrm{global}}}, \mathbf{W}^{K^{\mathrm{global}}}, \mathbf{W}^{V^{\mathrm{global}}}$ are learnable matrices.

**Map Attention and Output.** The map attention block is a multi-head attention layer with keys and values from the encoded agent-centric vectorized map (as in [34], we encode the map via an attention layer which transforms waypoints associated with each lane into a lane vector) and captures the interaction between agents and map. The map attention is applied to each agent independently as the map is agent-centric. The output encoded trajectory is projected back to the input dimension $d_s + d_a$ and results in the predicted clean action trajectory $\hat{\tau}_a^0$. At test time, we additionally apply iterative guidance with a differentiable loss function $\mathcal{J}$ (see Section 3.4 and Appendix A.2) on the predicted action trajectory. Finally, we apply dynamics to get the predicted state trajectory.

## 3.4 Guided Generation with Language

A language interface for the powerful diffusion model would enable the user to easily control trajectories with minimum domain knowledge. However, the absence of paired text-to-traffic data renders direct training of such a model infeasible. Hence, we explore using an intermediary representation to bridge the two. Recent advancements in LLMs facilitate high-quality conversion from natural language commands into code. Meanwhile, the diffusion model exhibits control over its generation through guidance from a loss function. Thus, we suggest utilizing a loss function implemented in code to bridge the two. Since the guidance loss function must operate on trajectories, we provide helper functions for coordinate transformations and a handful of text-loss function paired examples alongside the user's query to the LLM. We then utilize the returned loss function to guide the diffusion model, as discussed in Section 3.2, for generating query-compliant traffic simulation.

**Guidance Formulation.** Building upon prior work [13, 6], we apply *guidance* to sampled trajectories from our diffusion model at each denoising step to satisfy a predefined objective. Guidance uses the gradient of the loss $\mathcal{J}$ to perturb the model's predicted mean such that each denoising step (in Equation (2)) becomes: $p_\theta(\boldsymbol{\tau}_a^{k-1} \mid \boldsymbol{\tau}^k, \mathbf{c}) \approx \mathcal{N}(\boldsymbol{\tau}_a^{k-1}; \boldsymbol{\mu} + \boldsymbol{\Sigma}^k \nabla_{\boldsymbol{\mu}} \mathcal{J}(\boldsymbol{\mu}), \boldsymbol{\Sigma}^k)$ (see Appendix A.2).

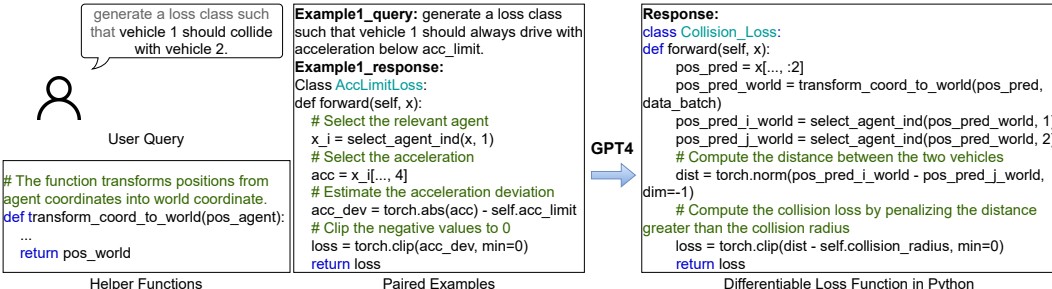

Figure 3: Example of prompting and querying LLM for a loss function promoting two vehicles to collide.

**Language Guidance.** Rather than training a classifier or reward function [13, 12] or defining an analytical reward function [6] for $\mathcal{J}$, we use GPT4 [39] to translate the intention in a user language query into the corresponding guidance function. In particular, we pass a few helper functions used to manipulate trajectory coordinates, a couple of (query → loss function code) paired examples, and the user query into GPT4, and extract an implemented loss function from the returned message. Figure 4 shows an example of querying for a loss function that causes a vehicle to collide with another one. The input paired example shows GPT4 how to apply a sequence of differentiable operations to generate a loss with respect to the predicted trajectory ("x"). We also provide a list of helper functions for trajectory manipulation (such as coordinate transformations) for GPT4 to leverage as we have found these can help to avoid minor mistakes on common operations (see Appendix F.2 for additional helper functions and pair examples). GPT4 returns a loss function that penalizes trajectories for not having a collision between the specified vehicles, which will guide the diffusion model to generate trajectories having such a collision. The guided sampling is performed jointly for all the agents in a scene.

**Traffic Simulation.** We perform closed-loop traffic simulation of each scene for 10 seconds. In particular, we apply our model for all the agents in a standard control loop: at each step, the model generates a guided trajectory and takes the first few actions before re-planning at 2 Hz.

## 4  Experiments

Following the setup (Section 4.1), we conduct experiments to affirm: CTG++ can effectively produce realistic and query-compliant traffic behaviors (Section 4.2), and compared with strong baselines, CTG++ yields superior trade-offs among stability, rule satisfaction, and realism (Section 4.3).

### 4.1  Experimental Setup

**Dataset.** nuScenes [19] is an extensive real-world driving dataset encompassing 5.5 hours of precise trajectories from two cities, featuring diverse scenarios and heavy traffic. We train all models using the training split and evaluate them on 100 scenes randomly selected from the validation split. Our focus is exclusively on simulating moving vehicles, as they are the most control-relevant entities.

**Metrics.** Following [6, 5], we evaluate stability (*i.e.*, avoiding collisions and off-road driving), controllability, and realism of generated trajectories. We evaluate *stability* by reporting the failure rate (**fail**), measured as the percentage of agents encountering a collision or road departure in a scene. We evaluate *controllability* using rule-specific violation metrics (**rule**) (see Appendix E.2). To assess *realism*, we compare data statistics between generated trajectories and ground truth trajectories from the dataset by calculating the Wasserstein distance between their normalized histograms of driving profiles. We measure *realism* using realism deviation (**real**), which is the average of the distribution distance for the longitudinal acceleration magnitude, lateral acceleration magnitude, and jerk. We introduce a new scene-level realism metric (**rel real**) which is the average of the distribution distance for relative (averaged over every pair of vehicles in a scene) longitudinal acceleration magnitude, relative lateral acceleration magnitude, and relative jerk.

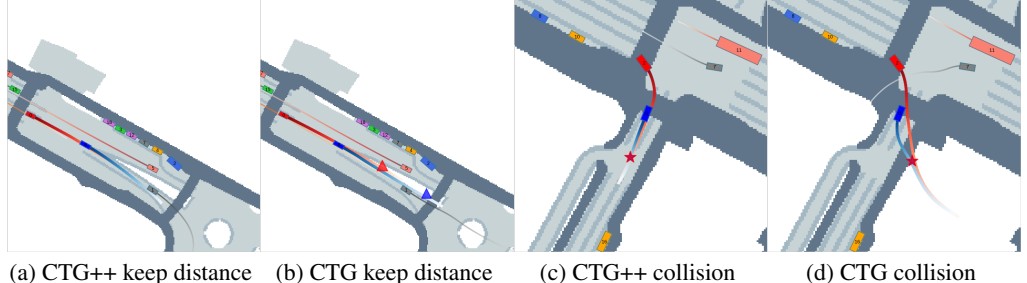

| (a) CTG++ keep distance | (b) CTG keep distance | (c) CTG++ collision | (d) CTG collision |

Figure 4: Generated trajectories for query "vehicle A should always keep within 10-30m from vehicle B" and "vehicle A should collide with vehicle B", respectively. The collision and offroad locations are marked in ☆ and △. For both CTG++ and CTG, our language interface generate query compliant trajectories. However, CTG++ does not sacrifice other aspects like keeping on-road and smoothness while CTG does.

**Traffic Model Baselines.** The closest related work on rule-compliant traffic generation is **CTG** [6], a traffic model based on conditional diffusion. We also consider **BITS** [5], a bi-level imitation learning model, and adapt its sampling ranking function to use our loss function. Its variant, **BITS+opt**, applies optimization to the output action trajectory for controllability. To ensure a fair comparison, this optimization employs the same loss function as the one used for guidance in CTG++.

**Large Language Model.** We use OpenAI's *GPT-4* [39] (accessed through the OpenAI API) for evaluating the language interface. We do not train the LLM and use only few-shot prompting.

## 4.2 Case Study of Language Interface

We conduct two case studies on queries for common traffic behaviors demonstrating CTG++ can effectively generate trajectories satisfying user's queries. Both queries involve the interaction of two vehicles. The first query "vehicle A should always keep within 10-30m from vehicle B" (*GPT keep distance*) is a common scene in the real world and the generated scene can be used to test vehicle following. Figure 4a shows the simulation generated by CTG++: vehicle A (in red) follows vehicle B (in blue) to go straight and slightly turn right with a safe distance as specified. Both vehicles along with the background vehicles have smooth motion without critical failures during the rollout. The second query "vehicle A should collide with vehicle B" (*GPT collision*) makes it possible to potentially generate scenarios like those from a crash report [7] and test a vehicle's reaction in dangerous situations. Figure 4c shows the simulation generated by CTG++: vehicle A collides with vehicle B while turning right, as desired. The motion of all vehicles is smooth and does not have collisions other than the one requested. The generated query-compliant trajectories for both cases show that our proposed language interface in CTG++ enables effective text-to-traffic generation.

## 4.3 Evaluation of Traffic Model

Table 1: Quantitative results of CTG++ and the baselines under GPT-generated rules and STL rules.

| | GPT keep distance | | | | GPT collision | | | | no collision | | | | speed limit | | | |
| | fail | rule | real | rel real | fail | rule | real | rel real | fail | rule | real | rel real | fail | rule | real | rel real |
|---|---|---|---|---|---|---|---|---|---|---|---|---|---|---|---|---|
| BITS | 0.183 | 2.615 | 0.116 | 0.362 | **0.176** | 0.660 | 0.107 | 0.359 | 0.092 | 0.065 | 0.099 | 0.352 | 0.111 | 0.559 | 0.104 | 0.352 |
| BITS+opt | 0.240 | **0.000** | 0.097 | 0.360 | 0.277 | 0.130 | **0.068** | 0.362 | 0.109 | 0.041 | 0.070 | 0.353 | 0.162 | 0.120 | 0.058 | 0.353 |
| CTG | 0.343 | **0.000** | **0.077** | 0.342 | 0.356 | **0.000** | 0.074 | 0.349 | 0.142 | 0.052 | 0.044 | 0.346 | 0.128 | 0.029 | 0.075 | 0.350 |
| CTG++ | **0.173** | **0.000** | **0.077** | **0.331** | 0.264 | **0.000** | 0.085 | **0.331** | **0.084** | **0.036** | **0.040** | **0.332** | **0.083** | **0.028** | **0.043** | **0.344** |

| | target speed | | | | no offroad | | | | goal waypoint+target speed | | | | | stopregion+offroad | | | | |
| | fail | rule | real | rel real | fail | rule | real | rel real | fail | rule1 | rule2 | real | rel real | fail | rule1 | rule2 | real | rel real |
|---|---|---|---|---|---|---|---|---|---|---|---|---|---|---|---|---|---|---|
| BITS | 0.111 | 1.526 | 0.114 | **0.355** | 0.097 | 0.018 | 0.099 | 0.355 | 0.111 | 2.261 | 1.010 | 0.115 | 0.358 | 0.121 | 0.005 | 1.690 | 0.068 | 0.353 |
| BITS+opt | 0.257 | 0.742 | **0.072** | 0.356 | 0.105 | 0.005 | 0.100 | 0.358 | 0.254 | 3.681 | 0.746 | 0.079 | 0.342 | 0.095 | 0.020 | 2.053 | 0.097 | 0.354 |
| CTG | 0.091 | 0.281 | 0.105 | 0.379 | 0.172 | **0.002** | 0.042 | 0.346 | 0.118 | 2.388 | **0.387** | 0.052 | 0.345 | 0.128 | **0.002** | 0.808 | **0.040** | 0.336 |
| CTG++ | **0.060** | **0.274** | 0.082 | 0.370 | **0.097** | 0.004 | **0.038** | **0.328** | **0.101** | 2.352 | 0.396 | **0.038** | **0.338** | **0.081** | 0.003 | **0.411** | 0.076 | **0.324** |

We assess the traffic model component of CTG++ under the two GPT-generated rules (as described in Section 4.2) and six STL rules from [6] (see Appendix E.1 for details of each rule). The quantitative results in Table 1 underscore CTG++'s superiority over baselines with a good balance between stability, rule satisfaction, and realism. Specifically, CTG++ secures the lowest failure rate and scene-level realism deviation in 7 out of 8 settings, reflecting its effective scene-level modeling and

enhanced interaction dynamics. Furthermore, CTG++ also achieves the lowest rule violation and agent-level realism deviation for the majority of settings, demonstrating that enhanced interaction modeling does not compromise agent-level realism or rule adherence.

Table 2: Ablation study of CTG++ features.

|  | fail | rule | real | rel real |
|---|---|---|---|---|
| CTG++ | **0.173** | **0.000** | **0.077** | **0.331** |
| CTG++ no edge | 0.227 | **0.000** | **0.077** | 0.341 |
| CTG++ scene | 0.886 | 1.043 | 0.127 | 0.392 |

Qualitatively, CTG++ generates rule-compliant trajectories featuring more realistic motion with fewer instances of collisions or off-road incidents than the baselines. We provide examples of CTG-generated trajectories (when using the same language generated loss functions) for the same scenes as those previously shown for CTG++ (Figure 4b and Figure 4d). Specifically, in Figure 4b, CTG's trajectories display off-road instances (indicated by △) involving two vehicles and a background vehicle. Similarly, in Figure 4d, for CTG's trajectories, the collision between vehicles A and B (in ☆) occurs off-road, and the background vehicle has a curvy, unrealistic path. See Appendix C for qualitative comparisons under STL rules.

**Ablation Study.** We evaluate the efficacy of our spatial module and the utilization of agent coordinates. As demonstrated in Table 2, CTG++ surpasses **CTG++ no edge** (i.e., CTG++ but having $\mathbf{e}_t^{ij}$ in Equation (3) replaced by zeros) and **CTG++ scene** (i.e., CTG++ but using scene-centric coordinates) under the *GPT Keep Distance* rule. The absence of edge information leads to increased failure rates due to decreased awareness of interactions. Moreover, the use of scene-centric coordinates notably inflates failure rates and realism deviations, as the traffic rapidly deviates from the training distribution during the rollout. To visualize the ef-

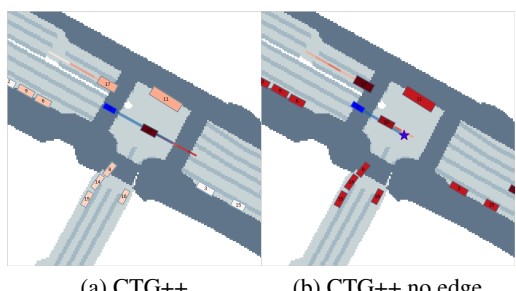

(a) CTG++                    (b) CTG++ no edge

Figure 5: Darker red means higher attention by the blue vehicle. Without edge information, CTG++ no edge results in a collision (in ☆).

fectiveness of spatial attention, we display the attention maps for a vehicle of interest (highlighted in blue) for both CTG++ and **CTG++ no edge** when guidance is not applied. As depicted in Figure 5, CTG++ guides the vehicle to pay attention to relevant neighboring agents, while **CTG++ no edge** results in arbitrary attention and a consequent collision (marked as ☆) with the vehicle ahead.

## 5   Conclusion

**Summary.** In this paper, we present, CTG++, a language-guided scene-level conditional diffusion model for realistic, query-compliant traffic simulation. In particular, we leverage an LLM for translating a user query into a differentiable loss function and propose a scene-level conditional diffusion model (with a spatial-temporal transformer architecture) to translate the loss function into realistic, query compliant trajectories. Extensive evaluation demonstrates the effectiveness of our approach.

**Limitations and Future Work.** CTG++ currently does not support complex commands involving many interactions with map (see Appendix F.4). Our framework can be extended by using a multi-modal LLM that takes in vision data (e.g., bird's-eye view map) for a finer control of the traffic and thus supports more complex commands. Second, our framework does not support automatic error detection and fixing for the GPT4 generated loss function. As the loss function (in code format) can have wrong semantics, it is necessary to instruct GPT4 to detect and repair it. Our framework can be potentially extended to provide the simulation running results to GPT4 to iteratively instruct it to fix the generated loss function. Third, current trajectory generation is relatively slow and take about 1 minute to generate each simulated scenario. Recent work which uses distillation [40] to greatly speed up the generation process can be leveraged to reduce the time cost. Our work opens up many possibilities including adapting our architecture to general multi-agent robotic tasks and using our proposed two-step approach for other tasks with no paired text-behavior data.

## Acknowledgment

The authors want to thank Sushant Veer and Shuhan Tan for valuable discussions. This work started when Ziyuan Zhong interned at NVIDIA Research. He is also supported by NSF CCF 1845893 and IIS 2221943.

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

# A  Algorithm of Training and Sampling in Details

We mostly follow the training and sampling procedures from [6] and show the detailed algorithms for training and sampling in the following.

## A.1  Training

**Algorithm 1** Training

---

1: **Require** a real-world driving dataset $D$, conditional diffusion model to train $\boldsymbol{\mu}_\theta^0$, transition function $f$, denoising steps $K$.
2: **while** not converge **do**
3:   $\mathbf{c}, \boldsymbol{\tau}^0 \sim D$
4:   $k \sim \{1, \ldots, K\}$
5:   $\epsilon \sim \mathcal{N}(\mathbf{0}, \boldsymbol{I})$
6:   Corrupt action trajectory $\boldsymbol{\tau}_a^k = \sqrt{\bar{\alpha}_k}\boldsymbol{\tau}_a^0 + \sqrt{1 - \bar{\alpha}_k}\epsilon$ with $\bar{\alpha}_k = \prod_{l=0}^k 1 - \beta_l$
7:   Get the corresponding state trajectory $\boldsymbol{\tau}_s^k = f(s_0, \boldsymbol{\tau}_a^k)$
8:   Use model to predict the uncorrupted trajectory $\hat{\boldsymbol{\tau}}_a^0 = \boldsymbol{\mu}_\theta^0(\boldsymbol{\tau}^k, k, \mathbf{c})$
9:   Get the predicted state trajectory $\hat{\boldsymbol{\tau}}^0 = [\hat{\boldsymbol{\tau}}_a^0; f(s_0, \hat{\boldsymbol{\tau}}_a^0)]$
10:   Take gradient step on $\nabla_\theta \|\boldsymbol{\tau}^0 - \hat{\boldsymbol{\tau}}^0\|^2$
11: **end while**

---

Contrary to [6], which samples trajectories at the agent level, we opt for scene-level trajectory sampling, allowing the model to make joint predictions on all scene agents. The process is detailed in Algorithm 1. During each training iteration, the context $\mathbf{c}$ and the ground-truth trajectory $\boldsymbol{\tau}^0$ are sampled from a real-world driving dataset, and the denoising step $k$ is uniformly selected from $\{1, \ldots, K\}$. We derive the noisy input $\boldsymbol{\tau}^k$ from $\boldsymbol{\tau}^0$ by initially corrupting the action trajectory via $\boldsymbol{\tau}_a^k = \sqrt{\bar{\alpha}_k}\boldsymbol{\tau}_a^0 + \sqrt{1 - \bar{\alpha}_k}\epsilon$, with $\epsilon \sim \mathcal{N}(0, \mathbf{I})$ and $\bar{\alpha}_k = \prod_{l=0}^k 1 - \beta_l$. Subsequently, the corresponding state is computed as $\boldsymbol{\tau}_s^k = f(s_0, \boldsymbol{\tau}_a^k)$. The diffusion model indirectly parameterizes $\boldsymbol{\mu}_\theta$ in eq. (2) by predicting the uncorrupted trajectory $\hat{\boldsymbol{\tau}}^0 = [\hat{\boldsymbol{\tau}}_a^0; f(s_0, \hat{\boldsymbol{\tau}}_a^0)]$, where $\hat{\boldsymbol{\tau}}_a^0 = \boldsymbol{\mu}_\theta^0(\boldsymbol{\tau}^k, k, \mathbf{c})$ is the network's direct output (see [12, 13, 41]). We then use a simplified loss function to train the model as follows:

$$L(\theta) = \mathbb{E}_{\epsilon, k, \boldsymbol{\tau}^0, \mathbf{c}}\left[\|\boldsymbol{\tau}^0 - \hat{\boldsymbol{\tau}}^0\|^2\right]. \tag{5}$$

A cosine variance schedule [13, 41] is utilized in the diffusion process, employing $K = 100$ diffusion steps.

## A.2  Sampling

We show the guided sampling algorithm in Algorithm 2 which is directly from [6] as the notations and procedure remain the same. The key difference is that our diffusion model formulation and backbone models are all at scene-level rather than agent-level as in [6]. The scene-level formulation helps to improve scene-level realism and decrease failure rates as the agents' interactions can be captured by the model inherently.

**Algorithm 2** Guided Sampling

---

1: **Require** conditional diffusion model $\boldsymbol{\mu}_\theta$, transition function $f$, guide $\mathcal{J}$, scale $\alpha$, covariances $\boldsymbol{\Sigma}^k$, diffusion steps $K$, inner gradient descent steps $W$, number of actions to take before re-planning $l$.
2: **while** not done **do**
3:   Observe state $s_0$ and context $\mathbf{c}$
4:   Initialize trajectory $\boldsymbol{\tau}_a^K \sim \mathcal{N}(\mathbf{0}, \boldsymbol{I})$; $\boldsymbol{\tau}_s^K = f(s_0, \boldsymbol{\tau}_a^K)$; $\boldsymbol{\tau}^K = [\boldsymbol{\tau}_a^K; \boldsymbol{\tau}_s^K]$
5:   **for** $k = K, \ldots, 1$ **do**
6:     $\boldsymbol{\mu} := \hat{\boldsymbol{\tau}}_a^{k-1} = \boldsymbol{\mu}_\theta(\boldsymbol{\tau}^k, k, \mathbf{c})$
7:     $\boldsymbol{\mu}^{(0)} = \boldsymbol{\mu}$
8:     **for** $j = 1, \ldots, W$ **do**
9:       $\boldsymbol{\mu}^{(j)} = \boldsymbol{\mu}^{(j-1)} + \alpha \nabla \mathcal{J}(\boldsymbol{\mu}^{(j-1)})$
10:       $\Delta\boldsymbol{\mu} = |\boldsymbol{\mu}^{(j)} - \boldsymbol{\mu}^{(0)}|$
11:       $\Delta\boldsymbol{\mu} \leftarrow \text{clip}(\Delta\boldsymbol{\mu}, -\beta_k, \beta_k)$
12:       $\boldsymbol{\mu}^{(j)} \leftarrow \boldsymbol{\mu}^{(0)} + \Delta\boldsymbol{\mu}$
13:     **end for**
14:     $\boldsymbol{\tau}_a^{k-1} \sim \mathcal{N}(\boldsymbol{\mu}^{(M)}, \boldsymbol{\Sigma}^k)$; $\boldsymbol{\tau}_s^{k-1} = f(s_0, \boldsymbol{\tau}_a^{k-1})$;
        $\boldsymbol{\tau}^{k-1} = [\boldsymbol{\tau}_a^{k-1}; \boldsymbol{\tau}_s^{k-1}]$
15:   **end for**
16:   Execute first $l$ actions of trajectory $\boldsymbol{\tau}_a^0$
17: **end while**

---

Following [6, 12], the predicted mean is a weighted sum between the predicted clean action trajectory and the input action trajectory from last denoising step:

$$\hat{\boldsymbol{\tau}}_a^{k-1} = \boldsymbol{\mu}_\theta(\boldsymbol{\tau}^k, k, \mathbf{c}) = \frac{\sqrt{\bar{\alpha}_{k-1}}\beta_k}{1 - \bar{\alpha}_k}\hat{\boldsymbol{\tau}}_a^0 + \frac{\sqrt{\alpha_k}\left(1 - \bar{\alpha}_{k-1}\right)}{1 - \bar{\alpha}_k}\boldsymbol{\tau}_a^k \tag{6}$$

The process of perturbing the predicted means from the diffusion model using gradients of a specified objective is summarized in algorithm 2. Following [6], we use an iterative projected gradient descent with the Adam optimizer and *filtration*, i.e., we guide several samples from the diffusion model and choose the one with the best rule satisfaction based on $\mathcal{J}$.

# B More Details on Architecture

## B.1 Detailed Architecture

We show the detailed data flow of our proposed architecture in Figure A1. Its main difference with the simplified architecture shown in Figure 2 is that we show position encoding, rFFN, and the details of the guidance module explicitly.

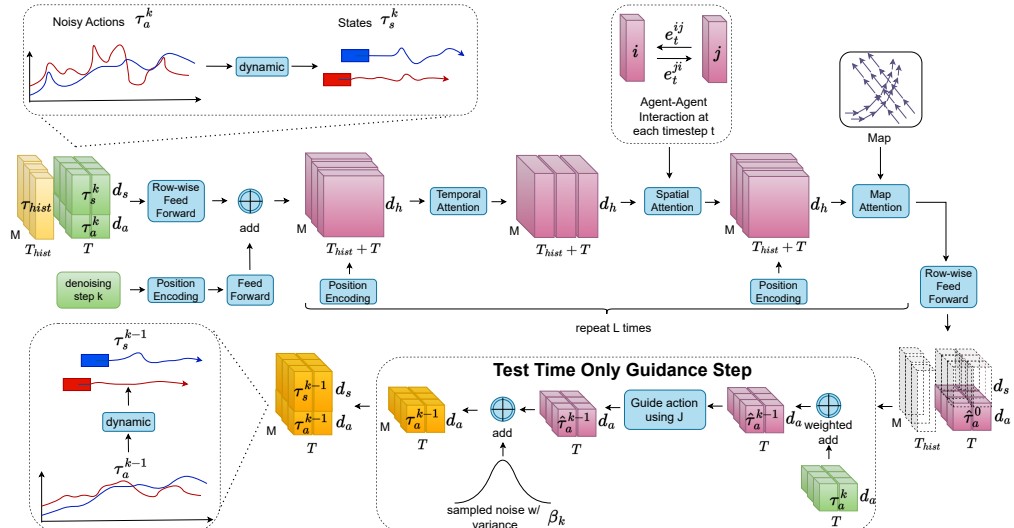

Figure A1: Test time denoising step using scene-level spatial-temporal transformer. $d_s$, $d_a$, and $d_h$ represent the dimensions of action, state, and latent for each vehicle per timestep.

## B.2 Gated Attention

Following [35], we use a variant of the original scaled dot-product attention block. In particular, we use a gating function to fuse the environmental features $m_i^t$ with the central agent's features $h_i^t$, enabling the block to have more control over the feature update. The resulting query, key, and value vectors of the social attention layer are taken as inputs to the block:

$$\begin{aligned}
\boldsymbol{\alpha}_t^i &= \text{softmax}\left(\frac{\mathbf{q}_t^{i\top}}{\sqrt{d_k}} \cdot \left[\left\{\mathbf{k}_t^{ij}\right\}_{j\in\mathcal{N}_i}\right]\right), \\
\mathbf{m}_t^i &= \sum_{j\in\mathcal{N}_i} \boldsymbol{\alpha}_t^{ij}\mathbf{v}_t^{ij}, \\
\mathbf{g}_t^i &= \text{sigmoid}\left(\mathbf{W}^{\text{gate}}\left[\mathbf{h}_t^i, \mathbf{m}_t^i\right]\right), \\
\hat{\mathbf{h}}_t^i &= \mathbf{g}_t^i \odot \mathbf{W}^{\text{self}}\mathbf{h}_t^i + \left(1 - \mathbf{g}_t^i\right) \odot \mathbf{m}_t^i,
\end{aligned} \tag{7}$$

where $\mathcal{N}_i$ is the set of agent i's neighbors (all the agents except the agent itself within a certain social radius), $\mathbf{W}^{\text{gate}}$ and $\mathbf{W}^{\text{self}}$ are learnable matrices, and denotes element-wise product $\odot$.

# C   Qualitative Comparison under STL rules

In this section, we show a few qualitative examples (Figure A2 - Figure A7) comparing CTG++ and the strongest baseline (in terms of rule satisfaction) under the STL rules. Overall, CTG++ generates realistic, rule-satisfying trajectories. The baseline method can usually also satisfy the rule. However, their trajectories usually sacrifice one or more of the following aspects: (1) the trajectories are curvy, unrealistic, (2) the trajectories involve off-road accidents, and (3) the agent interaction is sub-optimal leading to collision(s).

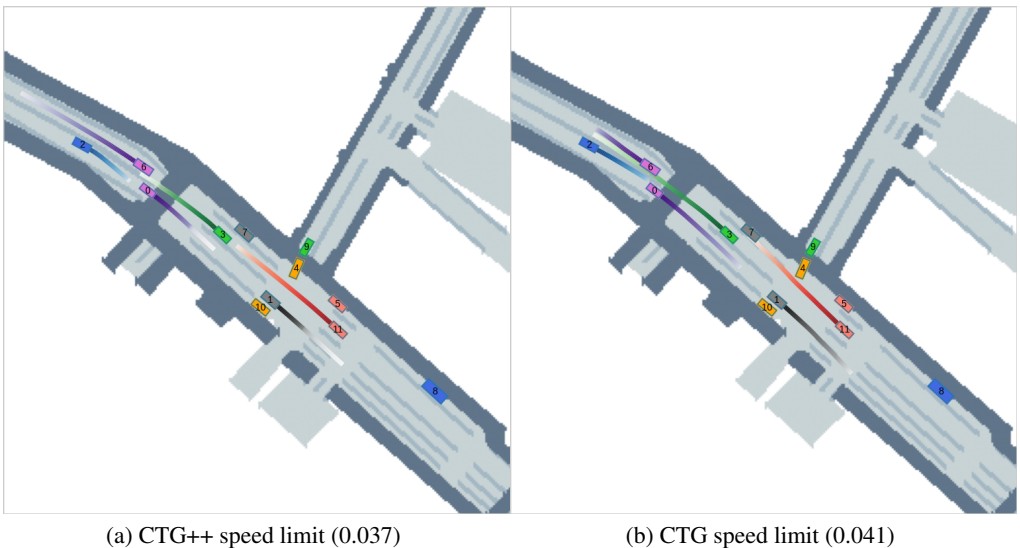

(a) CTG++ speed limit (0.037)            (b) CTG speed limit (0.041)

Figure A2: Qualitative comparison between CTG++ and CTG under speed limit STL rule (the numbers in parentheses represent rule violations). CTG++ achieves lower rule violation than CTG. Besides, CTG involves collision between the blue vehicle and the green vehicle.

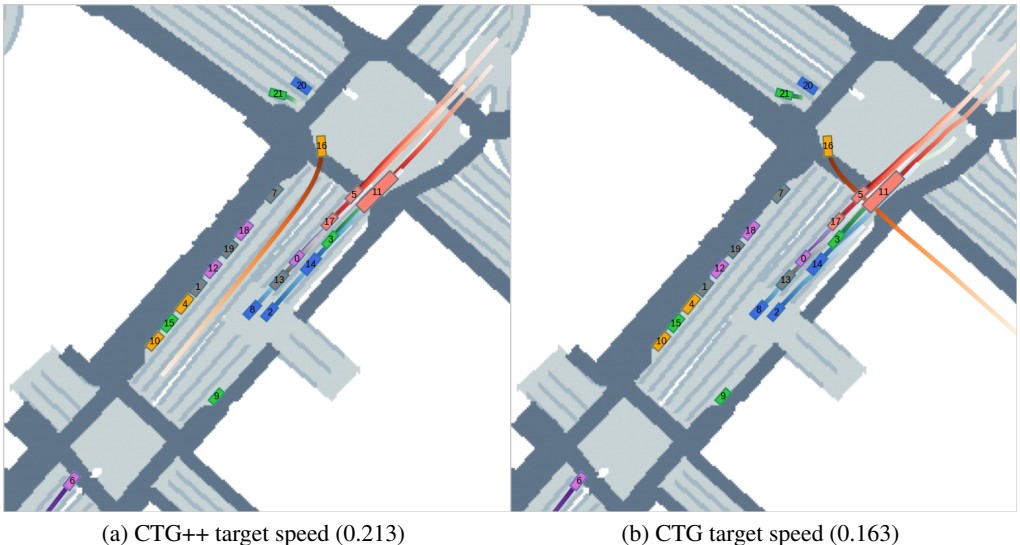

(a) CTG++ target speed (0.213)            (b) CTG target speed (0.163)

Figure A3: Qualitative comparison between CTG++ and CTG under target speed STL rule (the numbers in parentheses represent rule violations). Although CTG achieves a bit better target speed rule satisfaction, it involves a vehicle collides with crossing vehicles and then goes off-road.

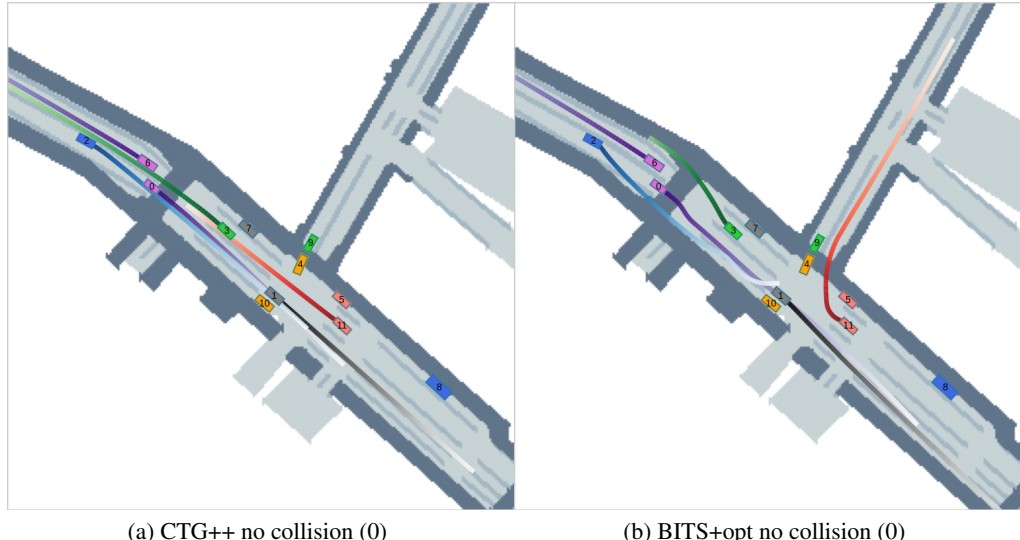

(a) CTG++ no collision (0)             (b) BITS+opt no collision (0)

Figure A4: Qualitative comparison between CTG++ and BITS+opt under no collision STL rule (the numbers in parentheses represent rule violations). Both methods satisfies the rule perfectly as no collision happens. However, BITS+opt have highly curvy, unrealistic trajectories as the cost of satisfying the rule.

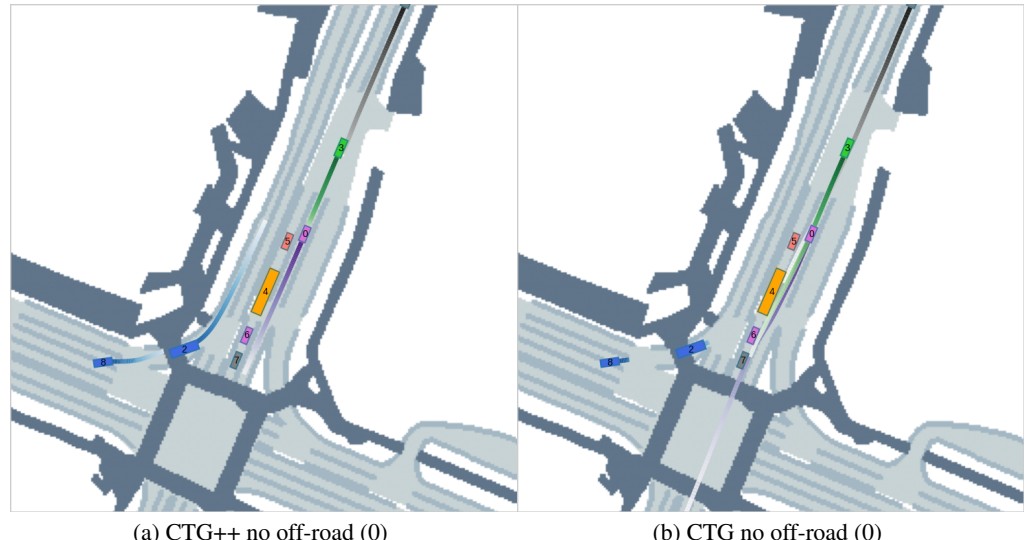

(a) CTG++ no off-road (0)             (b) CTG no off-road (0)

Figure A5: Qualitative comparison between CTG++ and CTG under no off-road STL rule (the numbers in parentheses represent rule violations). Both methods satisfies the rule perfectly as no off-road happens. However, CTG lead to multiple collisions among the pink vehicle and vehicles that are stationary.

# D Hyperparameters

## D.1 Training Hyperparameters

CTG++ is trained on a machine with Intel i9 12900 and NVIDIA GeForce RTX 3090. It takes approximately 10 hours to train CTG++ for 50K iterations. We use Adam optimizer with a learning rate of 1e-4.

## D.2 Pair Selection Criteria for GPT query based rules

We choose two vehicles A and B in each scene such that they satisfy the following criteria:

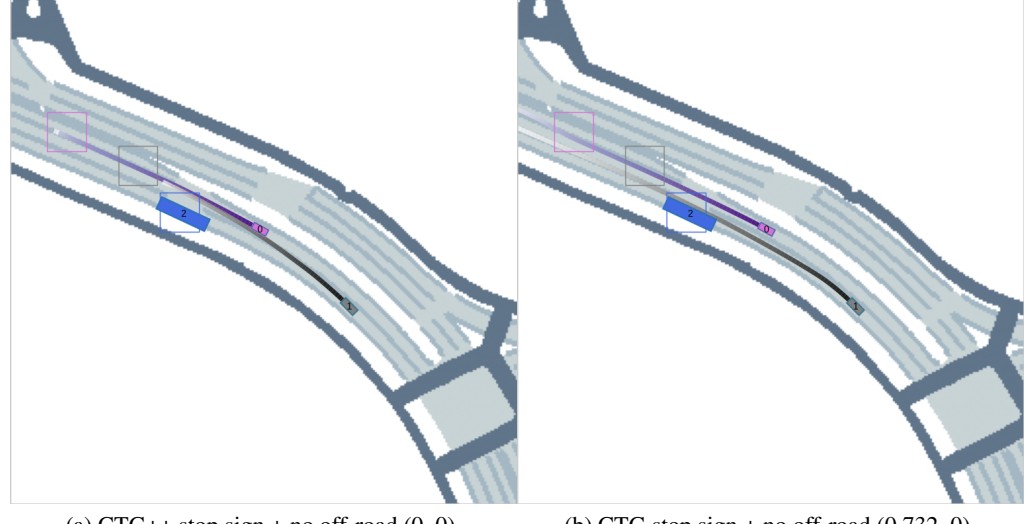

(a) CTG++ stop sign + no off-road (0, 0)  (b) CTG stop sign + no off-road (0.732, 0)

Figure A6: Qualitative comparison between CTG++ and CTG under stop sign and no off-road STL rule (the numbers in parentheses represent rules violations). Vehicles are supposed to stop within the marked bounding boxes without going off-road. CTG++ satisfies both rules while CTG only satisfies the no off-road rule. Besides, CTG involves a collision between the grey vehicle and the blue vehicle.

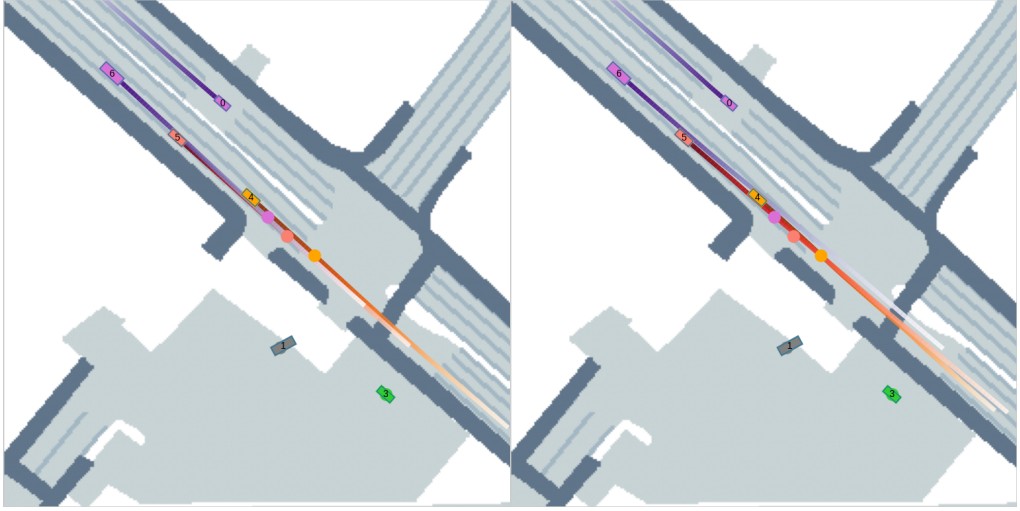

(a) CTG++ goal waypoint + target speed (0.991, 0.296)  (b) CTG goal waypoint + target speed (1.24, 0)

Figure A7: Qualitative comparison between CTG++ and CTG under goal waypoint + target speed STL rule (the numbers in parentheses represent rules violations). Vehicles are supposed to reach the marked waypoints with target speed (same speed as in the dataset). CTG++ satisfies both rules better than CTG. Besides, CTG involves a collision between the two orange vehicles in the end.

- Both have current speed larger than 2m/s.

- At 0s and 2s, the distance between A and B is within the range 10m to 30m.

- At 0s and 2s, the orientation difference between a and b is smaller than 108 degrees (for GPT collision) and 36 degrees (for GPT keep distance).

The criteria is a coarse-grained filtration for those pairs that are more likely to have keep distance / collision interactions in the original training dataset. If more than one pair in the scene satisfy the following criteria, we select the pair with smallest distance. If none of pairs in a scene satisfy the

following criteria, we skip the scene. After the filtrations, out of the 100 validation scenes, we have 50 scenes remained for **GPT collision** and 40 scenes for **GPT keep distance**.

# E    Experiment Details

## E.1    Quantitative Rules

We assess the traffic model component of CTG++ under both GPT-generated rules and STL rules from [6]: (1) *GPT Keep Distance* and (2) *GPT Collision*, as described in Section 4.2 (refer to Appendix D.2 for pair selection); (3) *No Collision* dictates that vehicles must avoid collisions with each other; (4) *Speed Limit* ensures vehicles do not exceed a set speed limit threshold (75% quantile of all moving vehicles in a given scene); (5) *Target Speed* requires vehicles to maintain a specified speed (50% of their speed in the ground truth scene) at each time step; (6) *No Offroad* prohibits vehicles from leaving the drivable area; (7) *Goal Waypoint and Target Speed* instructs vehicles to reach their designated goal while adhering to the specified target speeds; (8) *Stop Sign and No Off-road* requires vehicles to halt upon entering a stop sign region and avoid straying off the road.

## E.2    Metrics of Rule Violation

We provide the details for the metrics we use for measuring rule violation in this section. For all the metrics of rule violation, we average the metrics over all validation scenes. Besides, they are designed such that the smaller the better (i.e., rules are better satisfied).

**GPT Keep Distance.**  the following vehicle's (in the chosen pair) average L2 distance deviation from the specified range.

**GPT Collision.** if a collision happens between the two vehicles in the chosen pair.

**No Collision.** collision rate of all vehicles in a scene.

**Speed Limit.** average deviation from the speed limit of all vehicles in a scene.

**Target Speed.** average deviation from the target speed of all vehicles in a scene.

**No Offroad.**  off-road rate of all vehicles in a scene. We consider a vehicle going off-road if its center goes off-road.

**Goal Waypoint.** average vehicle's smallest l2 distance deviation from the specified corresponding goal waypoints of all vehicles in a scene.

**Stop Sign.** average smallest speed within the stop sign region of all vehicles in a scene.

# F    Details of Language Interface

In this section, we provide more details and limitation of our proposed language interface for traffic simulation.

## F.1    Details of Vehicle Indexing

In practice, instead of using color for vehicles which is used in Figure 1 for better illustration purpose, we use indices according to the context from the driving dataset. The user can tell GPT4 the vehicles to control via their indices, e.g., "vehicle 1 should collide with vehicle 2".

## F.2    Details of Prompting

In Figure 4, we provide an example of a pre-defined API function and a query-loss function pair. In our experiments, we additionally provide the following API functions:

**transform_coord_world_to_agent_i.** this function transform the predicted position and yaw from world coordinate to the agent i coordinate.

**select_agent_ind.** this function returns the slice of x with index i.

**get_current_lane_projection.** this function returns the projection of each vehicle predicted trajectory on its current lane in agent-centric coordinate.

**get_left_lane_projection.** this function is similar to get_current_lane except it returns the left lane waypoints. If there is no left lane, the original trajectory will be returned.

**get_right_lane_projection.** this function is similar to get_current_lane except it returns the right lane waypoints. If there is no right lane, the original trajectory will be returned.

In addition to the acceleration loss paired example shown in Figure 4, we provide another query-loss function pair example. "Generate a loss class such that, vehicle 1 should always stay on the left side of vehicle 2." The corresponding function penalize the cases when vehicle 1 not on the left size of vehicle 2. This function provides GPT4 a sense of the relationship between direction and the trajectories.

We additionally specify the dimension of the input trajectory and the input and output of the expected loss function wrapped in a loss class such that GPT4 know which dimension of the trajectory to operate on when needed: "The generated loss class should have a function: forward(x, data_batch, agt_mask). x is a tensor representing the current trajectory with shape (B, N, T, 6) where B is the number of vehicles (consisting of vehicle with index 0 to B-1), N is the number of samples for each vechile, T is the number of timesteps (each represents 0.1s), and 6 represents the (x, y, vel, yaw, acc, yawvel) in corresponding agent coordinate of each vehicle. data_batch is a dictionary that can be used as parameter for relevant APIs. The function should return a loss for every sample of every vehicle with the shape (B, N) or return a loss for every sample with the shape (N)."

### F.3 Success Examples with Complete Query and Response

In this section, we provide five generated success example programs and their queries. For all the examples, in addition to the user query, GPT4 is provided the APIs for the desired loss function and the helper functions as well as two paired examples: acceleration limit and stay on left. The examples help to elucidate two key aspects:

- the standard format of a loss class to generate.
- how to manipulate different trajectory dimensions using various helper functions (e.g., transform_coord_agents_to_world, transform_coord_world_to_agent_i, etc.) and construct a differentiable loss on the trajectory.

We first show the complete query (including system messages and user query sent to GPT4 API) and response for *GPT Collision* in Appendix F.3.1. Next, in Appendix F.3.2-Appendix F.3.5, for four other success exmples, we only show the user query and response as they use the same system messages as *GPT Collision* in Appendix F.3.1).

### F.3.1 Success Example: GPT Collision

The OpenAI GPT4 API allows one to specify multiple system messages (which includes the API of the loss function to generation, the APIs of the helper functions, and two paired prompted examples), and one user message (which is essentially the user query) (see [42] for more details). Thus, we list each system message and the user message in the following. The qualitative example of using this returned loss function has been shown in Figure 4c.

System Message 1 - Loss Function API and Helper Functions API:

```
"The generated loss class should have a function \
forward(x, data_batch, agt_mask). x is a tensor representing the
                                   current trajectory with shape (
```

B, N, T, 6) where B is the number of vehicles (consisting of vehicle 0 to B-1), N is the number of samples for each vechile, T is the number of timesteps (each represents 0.1s), and 6 represents the (x, y, vel, yaw, acc, yawvel) in corresponding agent coordinate of each vehicle. data_batch is a dictionary that can be used as parameter for relevant APIs. The function should return a loss for every sample of every vehicle with the shape (B, N) or return a loss for every sample with the shape (N). \

3    You can use PyTorch and the following APIs if needed amd you should not use other unseen functions: \

4    1. transform_coord_agents_to_world(pos_pred, yaw_pred, data_batch). pos_pred is the predicted position trajectory in agent coordinate with shape (B, N, T, 2) and 2 correspond to (x, y). yaw_pred is the predicted yaw trajectory in agent coordinate with shape (B, N, T, 1). The function transform the predicted position and yaw from their agent coordinates into the world coordinate. The function returns position and yaw in the world coordinate with the shape (B, N, T, 2) and (B, N, T, 1). \

5    2. transform_coord_world_to_agent_i(pos_pred_world, yaw_pred_world, data_batch, ind_k). pos_pred is the predicted position trajectory in world coordinate with shape (B, N, T, 2) and 2 represents (x, y). yaw_pred is the predicted yaw trajectory in world coordinate with shape (B, N, T, 1). data_batch is the dictionary mentioned before. ind_k is the index whose agent coordinate will be converted to. The function transform the predicted position and yaw from world coordinate to the agent i coordinate. The function returns position and yaw in the agent i coordinate with the shape (B, N, T, 2) and (B, N, T, 1). \

6    3. select_agent_ind(x, i). x has shape (B, N, T, k) where k can be any positive integer and i is a non-negative integer representing the selected index. This function returns the slice of x with index i with shape (N, T, k). \

7    4. get_current_lane_projection(pos_pred, yaw_pred, data_batch). pos_pred and yaw_pred have

System Message 2 - Paired Example 1 Query:

```
"Generate a loss class such that vehicle 1 should always drive with
                        acceleration below acc_limit."
```

System Message 2 - Paired Example 1 Response:

```
class AccLimitLoss(GuidanceLoss):
    '''
    Keep accelerations below a certain limit.
    '''
    def __init__(self, acc_limit):
        '''
        - acc_limit : acceleration limit.
        '''
        super().__init__()
        self.acc_limit = acc_limit

    def forward(self, x, data_batch, agt_mask=None):
        '''
        - x : the current trajectory (B, N, T, 6) where N is the
                                number of samples and 6 is
                                (x, y, vel, yaw, acc,
                                yawvel)

        - loss: (B, N)
        '''
        if agt_mask is not None:
            x = x[agt_mask]
        acc = x[..., [4]]
        acc_dev = torch.abs(acc) - self.acc_limit
        acc_loss = torch.clip(acc_dev, min=0)
        loss = torch.mean(acc_loss, dim=[-2, -1])

        return loss
```

System Message 3 - Paired Example 2 Query:

```
"Generate a loss class such that vehicle 20 should always stay on the
                            left side of vehicle 13."
```

System Message 3 - Paired Example 2 Response:

```python
class StayOnLeftLoss(GuidanceLoss):
    '''
    Vehicle with index target_ind should always keep on the left side
                                        of vehicle with index ref_ind.
    '''
    def __init__(self, target_ind=20, ref_ind=13, decay_rate=0.9):
        super().__init__()
        self.target_ind = target_ind
        self.ref_ind = ref_ind
        self.decay_rate = decay_rate

    def forward(self, x, data_batch, agt_mask=None):
        B, N, T, _ = x.shape
        if agt_mask is not None:
            x = x[agt_mask]
        # Select positions
        # (B,N,T,6) -> (B,N,T,2)
        pos_pred = x[..., :2]
        # Select yaws
        # (B,N,T,6) -> (B,N,T,1)
        yaw_pred = x[..., 3:4]
        # convert prediction from the respective agent coordinates to
                                                the world coordinate
        # (B,N,T,2), (B,N,T,1), dict -> (B,N,T,2), (B,N,T,1)
        pos_pred_world, yaw_pred_world =
        transform_coord_agents_to_world(pos_pred, yaw_pred, data_batch
                                        )
        # convert prediction from the world coordinate to the agent
                                                self.ref_ind coordinate
        # (B,N,T,2), (B,N,T,1), dict, int -> (B,N,T,2), (B,N,T,1)
        pos_pred_in_ref_ind, _ = transform_coord_world_to_agent_i(
                                        pos_pred_world,
                                        yaw_pred_world, data_batch,
                                        self.ref_ind)

        # select the relevant agents with index self.target_ind in the
                                                agent self.ref_ind
                                                coordinate
        # (B, N, T, 2), int -> (N, T, 2)
        pos_pred_i_in_ref_ind = select_agent_ind(pos_pred_in_ref_ind,
                                        self.target_ind)
        # select the relevant agents with index self.target_ind in the
                                                agent self.ref_ind
                                                coordinate
        # (B, N, T, 2), int -> (N, T, 2)
        pos_pred_j_in_ref_ind = select_agent_ind(pos_pred_in_ref_ind,
                                        self.ref_ind)

        # Since we only care about the y-axis, we only need to compute
                                                the y-axis (rather than
                                                both x and y axis)
                                                deviation.
        # (N, T, 2), (N, T, 2) -> (N, T)
        pos_dev = pos_pred_j_in_ref_ind[...,1] - pos_pred_i_in_ref_ind
                                        [...,1]

        # Clip the position deviation to 0 so that we only penalize
                                                the positive deviation
        # (N, T) -> (N, T)
        pos_loss = torch.clip(pos_dev, min=0)

        # penalize early steps more than later steps
```

```
45          exp_weights = torch.tensor([self.decay_rate ** t for t in
                                                    range(T)], device=x.device)
46          exp_weights /= exp_weights.sum()
47          pos_loss = pos_loss * exp_weights[None, :]
48          # Take the mean over time
49          # (N, T) -> (N)
50          pos_loss = pos_loss.mean(-1)
51
52          return pos_loss
```

User Message - User Query:

```
1  "Generate a loss class such that vehicle 1 should collide with vehicle
                                          2."
```

Received Message (from GPT4):

```
1  class CollisionLoss(GuidanceLoss):
2      '''
3      Vehicle with index target_ind should collide with vehicle with
                                              index ref_ind.
4      '''
5      def __init__(self, target_ind=1, ref_ind=2, collision_radius=1.0):
6          super().__init__()
7          self.target_ind = target_ind
8          self.ref_ind = ref_ind
9          self.collision_radius = collision_radius
10
11      def forward(self, x, data_batch, agt_mask=None):
12          B, N, T, _ = x.shape
13          if agt_mask is not None:
14              x = x[agt_mask]
15          # Select positions
16          # (B,N,T,6) -> (B,N,T,2)
17          pos_pred = x[..., :2]
18          # Select yaws
19          # (B,N,T,6) -> (B,N,T,1)
20          yaw_pred = x[..., 3:4]
21          # convert prediction from the respective agent coordinates to
                                              the world coordinate
22          # (B,N,T,2), (B,N,T,1), dict -> (B,N,T,2), (B,N,T,1)
23          pos_pred_world, yaw_pred_world =
24          transform_coord_agents_to_world(pos_pred, yaw_pred, data_batch
                                              )
25
26          # select the relevant agents with index self.target_ind in the
                                              world coordinate
27          # (B, N, T, 2), int -> (N, T, 2)
28          pos_pred_i_world = select_agent_ind(pos_pred_world, self.
                                              target_ind)
29          # select the relevant agents with index self.ref_ind in the
                                              world coordinate
30          # (B, N, T, 2), int -> (N, T, 2)
31          pos_pred_j_world = select_agent_ind(pos_pred_world, self.
                                              ref_ind)
32
33          # Compute the distance between the two vehicles
34          # (N, T, 2), (N, T, 2) -> (N, T)
35          dist = torch.norm(pos_pred_i_world - pos_pred_j_world, dim=-1)
36
37          # Compute the collision loss by penalizing the distance
                                              greater than the collision
                                              radius
38          # (N, T) -> (N, T)
```

```
39            collision_loss = torch.clip(self.collision_radius - dist, min=
                                                  0)
40
41            # Take the mean over time
42            # (N, T) -> (N)
43            loss = collision_loss.mean(-1)
44
45            return loss
```

The resulted loss function from GPT4 showcases its capabilities:

- It adeptly follows the template provided in the examples, producing a syntactically correct program.

- It comprehends the "collision" concept from the user query and knows how to integrate this concept with the input trajectory, as the examples we used for prompting (acceleration limit and stay on left) are not related to collisions at all. This understanding results in a loss that encourages trajectory collision between two vehicles, achieved through the helper functions.

### F.3.2   Success Example: GPT KeepDistance

In this success example, the returned program indicates that GPT4 understands the meaning of "keep within" and generates a function that promotes vehicle 1 to stay within certain distance of vehicle 2.

User Message - User Query:

```
1   "Generate a loss class such that vehicle 1 should always keep within
                                  10-30m from vehicle 2."
```

Received Message (from GPT4):

```
1   class KeepDistanceLoss(GuidanceLoss):
2       '''
3       Vehicle with index target_ind should always keep within
                                      min_distance and max_distance
                                      from vehicle with index ref_ind
                                      .
4       '''
5       def __init__(self, target_ind=1, ref_ind=2, min_distance=10,
                                      max_distance=30):
6           super().__init__()
7           self.target_ind = target_ind
8           self.ref_ind = ref_ind
9           self.min_distance = min_distance
10          self.max_distance = max_distance
11
12      def forward(self, x, data_batch, agt_mask=None):
13          B, N, T, _ = x.shape
14          if agt_mask is not None:
15              x = x[agt_mask]
16          # Select positions
17          # (B,N,T,6) -> (B,N,T,2)
18          pos_pred = x[..., :2]
19          # Select yaws
20          # (B,N,T,6) -> (B,N,T,1)
21          yaw_pred = x[..., 3:4]
22          # convert prediction from the respective agent coordinates to
                                          the world coordinate
23          # (B,N,T,2), (B,N,T,1), dict -> (B,N,T,2), (B,N,T,1)
24          pos_pred_world, yaw_pred_world =
25          transform_coord_agents_to_world(pos_pred, yaw_pred, data_batch
                                          )
```

```
26
27          # select the relevant agents with index self.target_ind in the
                                              world coordinate
28          # (B, N, T, 2), int -> (N, T, 2)
29          pos_pred_i_world = select_agent_ind(pos_pred_world, self.
                                              target_ind)
30          # select the relevant agents with index self.ref_ind in the
                                              world coordinate
31          # (B, N, T, 2), int -> (N, T, 2)
32          pos_pred_j_world = select_agent_ind(pos_pred_world, self.
                                              ref_ind)

34          # Compute the distance between the two vehicles
35          # (N, T, 2), (N, T, 2) -> (N, T)
36          distance = torch.norm(pos_pred_i_world - pos_pred_j_world, dim
                                              =-1)

38          # Compute the deviation from the desired distance range
39          # (N, T) -> (N, T)
40          distance_dev_min = self.min_distance - distance
41          distance_dev_max = distance - self.max_distance

43          # Clip the negative values to 0
44          # (N, T) -> (N, T)
45          distance_loss_min = torch.clip(distance_dev_min, min=0)
46          distance_loss_max = torch.clip(distance_dev_max, min=0)

48          # Combine the losses
49          # (N, T) -> (N, T)
50          distance_loss = distance_loss_min + distance_loss_max

52          # Take the mean over time
53          # (N, T) -> (N)
54          loss = torch.mean(distance_loss, dim=-1)

56          return loss
```

### F.3.3   Success Example: GPT SameDirection

In this success example, the returned program indicates that GPT4 understands the meaning of
"same direction" and generates a function that promotes vehicle 1 and vehicle 2 to move with the
same orientation.

User Message - User Query:

```
1  "Generate a loss class such that vehicle 1 should move along the same
                                      direction as vehicle 2."
```

Received Message (from GPT4):

```
1  class SameDirectionLoss(GuidanceLoss):
2      '''
3      Vehicle with index target_ind should move along the same direction
                                          as vehicle with index ref_ind.
4      '''
5      def __init__(self, target_ind=1, ref_ind=2):
6          super().__init__()
7          self.target_ind = target_ind
8          self.ref_ind = ref_ind

10     def forward(self, x, data_batch, agt_mask=None):
11         B, N, T, _ = x.shape
12         if agt_mask is not None:
13             x = x[agt_mask]
```

```
14          # Select yaws
15          # (B,N,T,6) -> (B,N,T,1)
16          yaw_pred = x[..., 3:4]
17          # convert prediction from the respective agent coordinates to
                                              the world coordinate
18          # (B,N,T,2), (B,N,T,1), dict -> (B,N,T,2), (B,N,T,1)
19          _, yaw_pred_world = transform_coord_agents_to_world(x[..., :2]
                                              , yaw_pred, data_batch)
20          # select the relevant agents with index self.target_ind
21          # (B, N, T, 1) -> (N, T, 1)
22          yaw_pred_i = select_agent_ind(yaw_pred_world, self.target_ind)
23          # select the relevant agents with index self.ref_ind
24          # (B, N, T, 1) -> (N, T, 1)
25          yaw_pred_j = select_agent_ind(yaw_pred_world, self.ref_ind)
26
27          # Compute the yaw deviation
28          # (N, T, 1), (N, T, 1) -> (N, T)
29          yaw_dev = torch.abs(yaw_pred_i - yaw_pred_j).squeeze(-1)
30
31          # Normalize the yaw deviation to [0, pi]
32          # (N, T) -> (N, T)
33          yaw_dev = torch.fmod(yaw_dev, 2 * math.pi)
34          yaw_dev = torch.min(yaw_dev, 2 * math.pi - yaw_dev)
35
36          # Penalize deviations larger than a small threshold (e.g., 0.1
                                              radian)
37          threshold = 0.1
38          yaw_loss = torch.clip(yaw_dev - threshold, min=0)
39
40          # Take the mean over time
41          # (N, T) -> (N)
42          loss = torch.mean(yaw_loss, dim=-1)
43
44          return loss
```

### F.3.4  Success Example: GPT Collision From Behind

In this success example, the returned program indicates that GPT4 understands the meaning of
"collide" and "from behind" and can create the respective losses (see collision_loss in line 49 and
not_behind_loss in line 53), respectively. It finally combines the two losses by taking their product
as the final loss (line 57).

User Message - User Query:

```
1   "Generate a loss class such that vehicle 1 should collide with vehicle
                                     2 from behind."
```

Received Message (from GPT4):

```
1   class CollideFromBehindLoss(GuidanceLoss):
2       '''
3       Vehicle with index target_ind should collide with vehicle with
                                              index ref_ind from behind.
4       '''
5       def __init__(self, target_ind=1, ref_ind=2, collision_threshold=1.
                                              0):
6           super().__init__()
7           self.target_ind = target_ind
8           self.ref_ind = ref_ind
9           self.collision_threshold = collision_threshold
10
11      def forward(self, x, data_batch, agt_mask=None):
12          B, N, T, _ = x.shape
```

```
13          if agt_mask is not None:
14              x = x[agt_mask]
15          # Select positions
16          # (B,N,T,6) -> (B,N,T,2)
17          pos_pred = x[..., :2]
18          # Select yaws
19          # (B,N,T,6) -> (B,N,T,1)
20          yaw_pred = x[..., 3:4]
21          # convert prediction from the respective agent coordinates to
                                            the world coordinate
22          # (B,N,T,2), (B,N,T,1), dict -> (B,N,T,2), (B,N,T,1)
23          pos_pred_world, yaw_pred_world =
24          transform_coord_agents_to_world(pos_pred, yaw_pred, data_batch
                                            )
25          # convert prediction from the world coordinate to the agent
                                            self.ref_ind coordinate
26          # (B,N,T,2), (B,N,T,1), dict, int -> (B,N,T,2), (B,N,T,1)
27          pos_pred_in_ref_ind, _ = transform_coord_world_to_agent_i(
                                            pos_pred_world,
                                            yaw_pred_world, data_batch,
                                             self.ref_ind)
28
29          # select the relevant agents with index self.target_ind in the
                                             agent self.ref_ind
                                             coordinate
30          # (B, N, T, 2), int -> (N, T, 2)
31          pos_pred_i_in_ref_ind = select_agent_ind(pos_pred_in_ref_ind,
                                            self.target_ind)
32          # select the relevant agents with index self.target_ind in the
                                             agent self.ref_ind
                                             coordinate
33          # (B, N, T, 2), int -> (N, T, 2)
34          pos_pred_j_in_ref_ind = select_agent_ind(pos_pred_in_ref_ind,
                                            self.ref_ind)
35
36          # Compute the distance between the two vehicles along the x-
                                            axis (longitudinal)
37          # (N, T, 2), (N, T, 2) -> (N, T)
38          dist_x = pos_pred_j_in_ref_ind[..., 0] - pos_pred_i_in_ref_ind
                                            [..., 0]
39          # Compute the distance between the two vehicles along the y-
                                            axis (lateral)
40          # (N, T, 2), (N, T, 2) -> (N, T)
41          dist_y = pos_pred_j_in_ref_ind[..., 1] - pos_pred_i_in_ref_ind
                                            [..., 1]
42
43          # Compute the squared distance between the two vehicles
44          # (N, T), (N, T) -> (N, T)
45          dist_squared = dist_x ** 2 + dist_y ** 2
46
47          # Compute the collision loss by checking if the squared
                                            distance is less than the
                                            collision threshold
48          # (N, T) -> (N, T)
49          collision_loss = (dist_squared < self.collision_threshold).
                                            float()
50
51          # Penalize the cases where the target vehicle is not behind
                                            the reference vehicle
52          # (N, T) -> (N, T)
53          not_behind_loss = torch.clip(-dist_x, min=0)
54
55          # Combine the collision loss and not_behind_loss
56          # (N, T), (N, T) -> (N, T)
57          total_loss = collision_loss * not_behind_loss
```

```
58
59            # Take the mean over time
60            # (N, T) -> (N)
61            loss = torch.mean(total_loss, dim=-1)
62
63            return loss
```

### F.3.5  Success Example: Lane Following

In this success example, the returned program indicates that GPT4 can extrapolate the number of vehicles of a scenario (although in both provided examples only two vehicles are involved, GPT4 can generalize the loss function involving three vehicles). Besides, GPT4 knows how to use functions described in helper function APIs but not even used in the provided paired examples. In particular, it knows to use get_current_lane_projection to get the lane information in order to promote vehicles lane following behavior when it was asked to let the vehicles to "follow their current lanes".

User Message - User Query:

```
1  "Generate a loss class such that vehicle 1, vehicle 2, and vehicle 3
                                       all follow their current lanes."
```

Received Message (from GPT4):

```
1  class LaneFollowingLoss(GuidanceLoss):
2      '''
3      Vehicles with indices target_inds should follow their current
                                              lanes.
4      '''
5      def __init__(self, target_inds=[1, 2, 3]):
6          super().__init__()
7          self.target_inds = target_inds
8
9      def forward(self, x, data_batch, agt_mask=None):
10          B, N, T, _ = x.shape
11          if agt_mask is not None:
12              x = x[agt_mask]
13          # Select positions
14          # (B,N,T,6) -> (B,N,T,2)
15          pos_pred = x[..., :2]
16          # Select yaws
17          # (B,N,T,6) -> (B,N,T,1)
18          yaw_pred = x[..., 3:4]
19
20          # Get the current lane projection in agent coordinate
21          # (B,N,T,2), (B,N,T,1), dict -> (B,N,T,3)
22          lane_proj = get_current_lane_projection(pos_pred, yaw_pred,
                                              data_batch)
23
24          # Compute the deviation from the lane projection
25          # (B,N,T,2), (B,N,T,3) -> (B,N,T,2)
26          pos_dev = pos_pred - lane_proj[..., :2]
27          yaw_dev = yaw_pred - lane_proj[..., 2:3]
28
29          # Compute the squared loss for position and yaw deviation
30          # (B,N,T,2) -> (B,N,T)
31          pos_loss = torch.sum(pos_dev ** 2, dim=-1)
32          # (B,N,T,1) -> (B,N,T)
33          yaw_loss = torch.squeeze(yaw_dev ** 2, dim=-1)
34
35          # Combine position and yaw loss
36          # (B,N,T), (B,N,T) -> (B,N,T)
37          total_loss = pos_loss + yaw_loss
38
```

```
39        # Select the loss for the target vehicles
40        # (B,N,T), list -> (len(target_inds), N, T)
41        target_losses = [select_agent_ind(total_loss, ind) for ind in
                                            self.target_inds]
42
43        # Stack the losses for target vehicles
44        # list -> (len(target_inds), N, T)
45        target_losses = torch.stack(target_losses, dim=0)
46
47        # Take the mean over time
48        # (len(target_inds), N, T) -> (len(target_inds), N)
49        target_losses = torch.mean(target_losses, dim=-1)
50
51        # Take the mean over target vehicles
52        # (len(target_inds), N) -> (N)
53        loss = torch.mean(target_losses, dim=0)
54
55        return loss
```

## F.4  Failure Cases

The main limitation of the current language interface is on the complex interactions between the vehicles and the map. As we don't explicitly pass the map information into the language interface, it cannot handle commands involving heavy interaction with the map. For example, "vehicle A and B move to the rightmost lane one by one and then both turn right at the next intersection". However, we believe if one provides more helper functions (especially those interacting with the map) and more relevant examples to the language interface, LLM can handle such more complicated commands. Additionally, we observe GPT4 can also fail to understand some concepts properly (e.g., "cut in"). As the current work aims to provide early promising results on the feasibility of text-to-traffic, we leave a deeper exploration for future work.

### F.4.1  Failure Example : Move to Rightmost and Right Turn

We provide a failure example below. The system messages remain the same as in Appendix F.3.1. The generated loss function fails because it fails to consider "turn right at the next intersection". This can be potentially fixed by providing a couple of more helper functions (e.g., a function to get the next intersection and corresponding lane's information) and an additional example that interacts with intersection. In other words, our framework can potentially support such queries via extensions on helper functions and additional examples. As in the current work, we mainly want to show a framework of using LLM on language-guided realistic traffic generation, we leave a comprehensive study covering all kinds of complex queries for traffic generation as future work.

User Message - User Query:

```
1   "vehicle 1 and 2 move to the rightmost lane one by one and then both
                                        turn right at the next intersection
                                        ."
```

Received Message (from GPT4):

```
1   class RightTurnLoss(GuidanceLoss):
2       '''
3       Vehicle 1 and 2 should move to the rightmost lane one by one and
                                            then both turn right at the
                                            next intersection.
4       '''
5       def __init__(self, target_inds=[1, 2], decay_rate=0.9):
6           super().__init__()
7           self.target_inds = target_inds
8           self.decay_rate = decay_rate
```

```
 9
10      def forward(self, x, data_batch, agt_mask=None):
11          B, N, T, _ = x.shape
12          if agt_mask is not None:
13              x = x[agt_mask]
14          # Select positions
15          # (B,N,T,6) -> (B,N,T,2)
16          pos_pred = x[..., :2]
17          # Select yaws
18          # (B,N,T,6) -> (B,N,T,1)
19          yaw_pred = x[..., 3:4]
20
21          # Get the right lane projection
22          # (B, N, T, 2), (B, N, T, 1), dict -> (B, N, T, 3)
23          right_lane_proj = get_right_lane_projection(pos_pred, yaw_pred
                                              , data_batch)
24
25          # Compute the deviation between the predicted position and the
                                              right lane projection
26          # (B, N, T, 2), (B, N, T, 3) -> (B, N, T, 2)
27          pos_dev = pos_pred - right_lane_proj[..., :2]
28
29          # Compute the squared distance
30          # (B, N, T, 2) -> (B, N, T)
31          pos_dev_sq = torch.sum(pos_dev ** 2, dim=-1)
32
33          # Penalize early steps more than later steps
34          exp_weights = torch.tensor([self.decay_rate ** t for t in
                                              range(T)], device=x.device)
35          exp_weights /= exp_weights.sum()
36          pos_dev_sq = pos_dev_sq * exp_weights[None, None, :]
37
38          # Take the mean over time
39          # (B, N, T) -> (B, N)
40          pos_loss = pos_dev_sq.mean(-1)
41
42          # Select the loss for target vehicles
43          # (B, N), list -> (len(target_inds), N)
44          target_loss = torch.stack([select_agent_ind(pos_loss, i) for i
                                              in self.target_inds])
45
46          # Take the mean over target vehicles
47          # (len(target_inds), N) -> (N)
48          loss = target_loss.mean(0)
49
50          return loss
```

### F.4.2    Failure Example: Cut In if Behind and On The Left

In this failure example, the returned program indicates that GPT4 fails to understand "cut in" as it only promotes vehicle 1 to be ahead of vehicle 2 when it is behind and on the left. However, cut-in requires vehicle 1 to also move horizontally (i.e., along the y axis).

User Message - User Query:

```
1  "Generate a loss class such that vehicle 1 should cut in ahead of
                                  vehicle 2 if it is behind vehicle 2
                                  and on its left lane."
```

Received Message (from GPT4):

```
1  class CutInLoss(GuidanceLoss):
2      '''
```

```python
        Vehicle with index target_ind should cut in ahead of vehicle with
                                             index ref_ind if it is behind
                                             vehicle ref_ind and on its left
                                             lane.
        '''
    def __init__(self, target_ind=1, ref_ind=2, decay_rate=0.9):
        super().__init__()
        self.target_ind = target_ind
        self.ref_ind = ref_ind
        self.decay_rate = decay_rate

    def forward(self, x, data_batch, agt_mask=None):
        B, N, T, _ = x.shape
        if agt_mask is not None:
            x = x[agt_mask]
        # Select positions
        # (B,N,T,6) -> (B,N,T,2)
        pos_pred = x[..., :2]
        # Select yaws
        # (B,N,T,6) -> (B,N,T,1)
        yaw_pred = x[..., 3:4]
        # convert prediction from the respective agent coordinates to
                                             the world coordinate
        # (B,N,T,2), (B,N,T,1), dict -> (B,N,T,2), (B,N,T,1)
        pos_pred_world, yaw_pred_world =
        transform_coord_agents_to_world(pos_pred, yaw_pred, data_batch
                                             )
        # convert prediction from the world coordinate to the agent
                                             self.ref_ind coordinate
        # (B,N,T,2), (B,N,T,1), dict, int -> (B,N,T,2), (B,N,T,1)
        pos_pred_in_ref_ind, _ = transform_coord_world_to_agent_i(
                                             pos_pred_world,
                                             yaw_pred_world, data_batch,
                                              self.ref_ind)

        # select the relevant agents with index self.target_ind in the
                                             agent self.ref_ind
                                             coordinate
        # (B, N, T, 2), int -> (N, T, 2)
        pos_pred_i_in_ref_ind = select_agent_ind(pos_pred_in_ref_ind,
                                             self.target_ind)
        # select the relevant agents with index self.target_ind in the
                                             agent self.ref_ind
                                             coordinate
        # (B, N, T, 2), int -> (N, T, 2)
        pos_pred_j_in_ref_ind = select_agent_ind(pos_pred_in_ref_ind,
                                             self.ref_ind)

        # Compute the x-axis and y-axis deviation
        # (N, T, 2), (N, T, 2) -> (N, T)
        pos_dev_x = pos_pred_j_in_ref_ind[..., 0] -
                                             pos_pred_i_in_ref_ind[...,
                                             0]
        pos_dev_y = pos_pred_j_in_ref_ind[..., 1] -
                                             pos_pred_i_in_ref_ind[...,
                                             1]

        # Check if the target vehicle is behind and on the left lane
                                             of the reference vehicle
        behind_and_left = (pos_dev_x > 0) & (pos_dev_y > 0)

        # Penalize the target vehicle for not cutting in ahead of the
                                             reference vehicle
        cut_in_loss = torch.where(behind_and_left, pos_dev_x, torch.
                                             zeros_like(pos_dev_x))
```

```
46
47          # Penalize early steps more than later steps
48          exp_weights = torch.tensor([self.decay_rate ** t for t in
                                                    range(T)], device=x.device)
49          exp_weights /= exp_weights.sum()
50          cut_in_loss = cut_in_loss * exp_weights[None, :]
51          # Take the mean over time
52          # (N, T) -> (N)
53          cut_in_loss = cut_in_loss.mean(-1)
54
55          return cut_in_loss
```

## F.5    Qualitative comparison between GPT4 and GPT3.5

We next show two examples where we send the same message but use GPT3.5 rather than GPT4 as the LLM. We keep the queries to be exactly the same as in Appendix F.3.1 and Appendix F.3.2, respectively. The received messages are shown in the following. Both loss functions are correct but are more similar to the provided examples by containing code on applying a decay on loss over time dimension (e.g., see line 45-48 in Received Message (from GPT3.5)). These lines of code are almost the exactly ones (except the loss name) in a provided example (stay on left) but these code are not always necessary to be included in the loss functions.

### F.5.1    GPT Collision (GPT3.5)

User Message - User Query:

```
1  "Generate a loss class such that vehicle 1 should collide with vehicle
                              2."
```

Received Message (from GPT3.5):

```
1  class CollisionLoss(GuidanceLoss):
2      '''
3      Vehicle with index target_ind should collide with vehicle with
                                      index ref_ind.
4      '''
5      def __init__(self, target_ind=1, ref_ind=2, decay_rate=0.9,
                                      collision_distance=2.0):
6          super().__init__()
7          self.target_ind = target_ind
8          self.ref_ind = ref_ind
9          self.decay_rate = decay_rate
10         self.collision_distance = collision_distance
11
12     def forward(self, x, data_batch, agt_mask=None):
13         B, N, T, _ = x.shape
14         if agt_mask is not None:
15             x = x[agt_mask]
16         # Select positions
17         # (B,N,T,6) -> (B,N,T,2)
18         pos_pred = x[..., :2]
19         # Select yaws
20         # (B,N,T,6) -> (B,N,T,1)
21         yaw_pred = x[..., 3:4]
22         # convert prediction from the respective agent coordinates to
                                              the world coordinate
23         # (B,N,T,2), (B,N,T,1), dict -> (B,N,T,2), (B,N,T,1)
24         pos_pred_world, yaw_pred_world =
25         transform_coord_agents_to_world(pos_pred, yaw_pred, data_batch
                                              )
26         # convert prediction from the world coordinate to the agent
                                              self.ref_ind coordinate
```

```python
            # (B,N,T,2), (B,N,T,1), dict, int -> (B,N,T,2), (B,N,T,1)
            pos_pred_in_ref_ind, _ = transform_coord_world_to_agent_i(
                                         pos_pred_world,
                                         yaw_pred_world, data_batch,
                                         self.ref_ind)

            # select the relevant agents with index self.target_ind in the
                                        agent self.ref_ind
                                        coordinate
            # (B, N, T, 2), int -> (N, T, 2)
            pos_pred_i_in_ref_ind = select_agent_ind(pos_pred_in_ref_ind,
                                        self.target_ind)
            # select the relevant agents with index self.target_ind in the
                                        agent self.ref_ind
                                        coordinate
            # (B, N, T, 2), int -> (N, T, 2)
            pos_pred_j_in_ref_ind = select_agent_ind(pos_pred_in_ref_ind,
                                        self.ref_ind)

            # Compute the distance between the two agents
            # (N, T, 2), (N, T, 2) -> (N, T)
            dist = torch.norm(pos_pred_i_in_ref_ind -
                                        pos_pred_j_in_ref_ind, dim=
                                        -1)

            # Clip the distance to the collision distance
            # (N, T) -> (N, T)
            dist_loss = torch.clip(self.collision_distance - dist, min=0)

            # penalize early steps more than later steps
            exp_weights = torch.tensor([self.decay_rate ** t for t in
                                        range(T)], device=x.device)
            exp_weights /= exp_weights.sum()
            dist_loss = dist_loss * exp_weights[None, :]
            # Take the mean over time
            # (N, T) -> (N)
            dist_loss = dist_loss.mean(-1)

            return dist_loss
```

### F.5.2   GPT Keep Distance (GPT3.5)

User Message - User Query:

```
"Generate a loss class such that vehicle 1 should always keep within
                        10-30m from vehicle 2."
```

Received Message (from GPT3.5):

```python
class KeepDistanceLoss(GuidanceLoss):
    '''
    Vehicle with index target_ind should always keep within min_dist
                                and max_dist from vehicle with
                                index ref_ind.
    '''
    def __init__(self, target_ind=1, ref_ind=2, min_dist=10, max_dist=
                                30, decay_rate=0.9):
        super().__init__()
        self.target_ind = target_ind
        self.ref_ind = ref_ind
        self.min_dist = min_dist
        self.max_dist = max_dist
        self.decay_rate = decay_rate

```

```python
def forward(self, x, data_batch, agt_mask=None):
    B, N, T, _ = x.shape
    if agt_mask is not None:
        x = x[agt_mask]
    # Select positions
    # (B,N,T,6) -> (B,N,T,2)
    pos_pred = x[..., :2]
    # Select yaws
    # (B,N,T,6) -> (B,N,T,1)
    yaw_pred = x[..., 3:4]
    # convert prediction from the respective agent coordinates to
    #                                   the world coordinate
    # (B,N,T,2), (B,N,T,1) -> (B,N,T,2), (B,N,T,1)
    pos_pred_world, yaw_pred_world =
    transform_coord_agents_to_world(pos_pred, yaw_pred, data_batch
                                    )
    # convert prediction from the world coordinate to the agent
    #                                   self.ref_ind coordinate
    # (B,N,T,2), (B,N,T,1), dict, int -> (B,N,T,2), (B,N,T,1)
    pos_pred_in_ref_ind, _ = transform_coord_world_to_agent_i(
                                    pos_pred_world,
                                    yaw_pred_world, data_batch,
                                    self.ref_ind)

    # select the relevant agents with index self.target_ind in the
    #                                   agent self.ref_ind
    #                                   coordinate
    # (B, N, T, 2), int -> (N, T, 2)
    pos_pred_i_in_ref_ind = select_agent_ind(pos_pred_in_ref_ind,
                                    self.target_ind)
    # select the relevant agents with index self.target_ind in the
    #                                   agent self.ref_ind
    #                                   coordinate
    # (B, N, T, 2), int -> (N, T, 2)
    pos_pred_j_in_ref_ind = select_agent_ind(pos_pred_in_ref_ind,
                                    self.ref_ind)

    # Compute the distance between the two agents
    # (N, T, 2), (N, T, 2) -> (N, T)
    dist = torch.norm(pos_pred_j_in_ref_ind -
                                    pos_pred_i_in_ref_ind, dim=
                                    -1)

    # Compute the deviation from the desired distance range
    # (N, T) -> (N, T)
    dist_dev = torch.where(dist < self.min_dist, self.min_dist -
                                    dist, torch.where(dist >
                                    self.max_dist, dist - self.
                                    max_dist, torch.zeros_like(
                                    dist)))

    # penalize early steps more than later steps
    exp_weights = torch.tensor([self.decay_rate ** t for t in
                                    range(T)], device=x.device)
    exp_weights /= exp_weights.sum()
    dist_dev = dist_dev * exp_weights[None, :]
    # Take the mean over time
    # (N, T) -> (N)
    dist_loss = dist_dev.mean(-1)

    return dist_loss
```

# G    Quantitative Evaluation Results with Additional Runs

To substantiate the superior performance of our method, we conduct additional experiments, the results of which are presented in Table 3. In these experiments, we compare CTG++ against the strongest baseline, CTG, across three distinct runs with varying random seeds. The settings are the same as those for Table 1 and we take the average and standard deviation of the three runs. We highlight the better value only when it is significantly better than the other (i.e., if the values of the two methods differ by at least the sum of their standard deviations). In all eight settings, CTG++ significantly performs better than CTG in terms of failure rate and scene-level realism. CTG++ also tends to perform better than CTG in terms of rule satisfaction (winning 4 and tied on 6). In terms of realism, CTG++ is comparable to CTG (winning 3, losing 2, and tied on 3). Thus, the results suggest that CTG++ significantly performs better than the strongest baseline CTG++.

Table 3: Quantitative results (mean with standard deviation of three runs) of CTG++ and the strongest baselines CTG under GPT-generated rules and STL rules. We highlight the winning method that is significantly better than the other (i.e., if the values of the two methods differ by at least the sum of their standard deviations).

| | GPT keep distance | | | | GPT collision | | | |
|---|---|---|---|---|---|---|---|---|
| | fail | rule | real | rel real | fail | rule | real | rel real |
| CTG | $0.327 \pm 0.02$ | $0 \pm 0$ | $0.07 \pm 0.006$ | $0.343 \pm 0.003$ | $0.346 \pm 0.018$ | $0 \pm 0$ | $\mathbf{0.071 \pm 0.004}$ | $0.349 \pm 0.002$ |
| CTG++ | $\mathbf{0.171 \pm 0.002}$ | $0 \pm 0$ | $0.071 \pm 0.006$ | $\mathbf{0.334 \pm 0.003}$ | $\mathbf{0.264 \pm 0.013}$ | $0 \pm 0$ | $0.084 \pm 0.004$ | $\mathbf{0.336 \pm 0.006}$ |

| | no collision | | | | speed limit | | | |
|---|---|---|---|---|---|---|---|---|
| | fail | rule | real | rel real | fail | rule | real | rel real |
| CTG | $0.137 \pm 0.01$ | $0.048 \pm 0.003$ | $0.048 \pm 0.005$ | $0.346 \pm 0.002$ | $0.129 \pm 0.002$ | $0.029 \pm 0$ | $0.077 \pm 0.002$ | $0.353 \pm 0.003$ |
| CTG++ | $\mathbf{0.085 \pm 0.002}$ | $0.045 \pm 0.001$ | $0.047 \pm 0.007$ | $\mathbf{0.326 \pm 0.006}$ | $\mathbf{0.087 \pm 0.004}$ | $0.028 \pm 0$ | $\mathbf{0.042 \pm 0.003}$ | $\mathbf{0.34 \pm 0.004}$ |

| | target speed | | | | no offroad | | | |
|---|---|---|---|---|---|---|---|---|
| | fail | rule | real | rel real | fail | rule | real | rel real |
| CTG | $0.083 \pm 0.007$ | $0.281 \pm 0.001$ | $0.108 \pm 0.003$ | $0.379 \pm 0.002$ | $0.167 \pm 0.008$ | $0.003 \pm 0$ | $0.041 \pm 0.002$ | $0.343 \pm 0.003$ |
| CTG++ | $\mathbf{0.062 \pm 0.001}$ | $\mathbf{0.272 \pm 0.002}$ | $\mathbf{0.083 \pm 0.004}$ | $\mathbf{0.371 \pm 0.004}$ | $\mathbf{0.104 \pm 0.008}$ | $0.003 \pm 0$ | $0.044 \pm 0.006$ | $\mathbf{0.323 \pm 0.005}$ |

| | goal waypoint+target speed | | | | | stopregion+offroad | | | | |
|---|---|---|---|---|---|---|---|---|---|---|
| | fail | rule1 | rule2 | real | rel real | fail | rule1 | rule2 | real | rel real |
| CTG | $0.135 \pm 0.015$ | $2.407 \pm 0.016$ | $0.39 \pm 0.003$ | $0.052 \pm 0.002$ | $0.343 \pm 0.002$ | $0.128 \pm 0.01$ | $0.003 \pm 0.001$ | $0.795 \pm 0.017$ | $\mathbf{0.046 \pm 0.006}$ | $0.336 \pm 0.003$ |
| CTG++ | $\mathbf{0.103 \pm 0.002}$ | $\mathbf{2.361 \pm 0.021}$ | $0.394 \pm 0.003$ | $\mathbf{0.039 \pm 0.001}$ | $\mathbf{0.336 \pm 0.004}$ | $\mathbf{0.08 \pm 0.012}$ | $0.003 \pm 0$ | $\mathbf{0.44 \pm 0.051}$ | $0.076 \pm 0.001$ | $\mathbf{0.323 \pm 0.006}$ |

