# OpenReview forum: "Language-Guided Traffic Simulation via Scene-Level Diffusion"
_robot-learning.org/CoRL/2023/Conference — CoRL 2023 Oral_

### Official Review · Reviewer_eWu2 · 2023-07-11

**Confidence:** 2
**Originality:** Good
**Technical Quality:** Fair
**Clarity Of Presentation:** Good
**Impact:** 2

**Recommendation:**

Weak Reject: I recommend rejecting the paper, but will not argue for my recommendation if the majority of other reviewers have a different opinion.

**Review:**

I am not very familiar with the task setting of traffic simulation. My review is mainly on the technical method and assumes the experimental benchmark and baselines are standard and competitive.

The transformer-based neural model seems sensible. While it is interesting to see GPT4 can generate loss functions for guiding diffusion models, it seems the authors are evaluating 8 template-based different tasks. It is not entirely clear to me to what extent the tasks are diverse in terms of natural language instructions. It is also unclear how GPT4 would perform in scenarios not tested by the authors. It is hard for me to assess the significance of the proposed framework because it is hard to assess the model’s capability outside the 8 scenarios that the authors constructed.

Is the main claim of the paper on the superiority of its transformer-based diffusion model or language-conditional generation capability? If former, it seems to me a more comprehensive benchmarking (potentially without language) will be convincing. If latter, it seems a certain level of diversity in the language instruction is needed to establish evidence for GPT4’s capability to generate good loss function for diffusion guidance.


**Quality Of The Limitations Section:**

Additional details required

**Questions For Rebuttal:**


It seems the language-based guidance is only sensible for an actual traffic state. Applying this loss function to the denoising process where t is not 0 seems strange to me. Can the authors comment on this?

I don’t have a good grasp of the absolute values of the metrics in Table 1. Can the authors give error bars to the results in Table 1?


**Robotics Focus:**

Relevant but unlikely to deploy to hardware in near future

**Summary Of Paper:**

This paper proposes a natural-language-conditional diffusion model for generating traffic scenes. It presents a sophisticated workflow where a transformer is used as the diffusion mode while GPT4 is used to generate a loss function conditional on the language instruction. This loss function is then used to guide a transformer-based diffusion model sampling by adding a score vector point to the gradient direction of this language-conditional loss.

**Summary Of Recommendation:**

The significance of this paper appears unclear to me (therefore leaning toward rejection). This may be due to my lack of background in this topic (therefore low confidence).

---

### Official Review · Reviewer_J6rY · 2023-07-18

**Confidence:** 4
**Originality:** Very Good
**Technical Quality:** Good
**Clarity Of Presentation:** Very Good
**Impact:** 3

**Recommendation:**

Strong Accept: I recommend accepting the paper and will argue for my recommendation even if other reviewers hold a different opinion.

**Review:**

This work is well organized and well written. The proposed CTG++ is interesting and novel.

I am curious about how to choose two interesting vehicles. In Fig. 3, it is listed in the response of LLM based on the select_agent_ind. If the query is shown as the one in Fig. 3, how does the LLM know the distance of two vehicles without any scenario context information? I suggest the authors provide a whole example of the LLM guidance in detail.


**Quality Of The Limitations Section:**

Additional details required

**Questions For Rebuttal:**

What is the real query of the LLM?

**Robotics Focus:**

Highly relevant to robotics but no hardware experiments

**Summary Of Paper:**

CTG++ is a very interesting and meaningful work for the realistic and controllable traffic simulation area. Different from CTG, it uses a scene-level conditional diffusion model with agent-centric coordinates, leveraging a spatial-temporal transformer backbone. In addition, a language interface is adept at generating trajectories that align with user-defined rules in language, which can improve scenario controllability and the user's interactivity. The experimental results prove the effectiveness of the proposed CTG++.

**Summary Of Recommendation:**

CTG++ uses a conditional diffusion model at the scene level with agent-centered coordinates and a spatial-temporal transformer backbone. In addition, a language interface is adept at generating trajectories that align with user-defined rules in language, which can improve the scenario controllability and the user's interactivity. This work is well organized and has a certain novelty.  It is beneficial for the autonomous driving community if this work is open-sourced.

---

### Official Review · Reviewer_qk9Q · 2023-07-19

**Confidence:** 3
**Originality:** Good
**Technical Quality:** Good
**Clarity Of Presentation:** Good
**Impact:** 3

**Recommendation:**

Weak Accept: I recommend accepting the paper, but will not argue for my recommendation if the majority of other reviewers have a different opinion.

**Review:**

## Strengths

- Model architectures that take inspiration from several subfields. Captures the multi-agent aspects of traffic simulation well.
- Novel approach of generating differentiable loss functions with LLM. This allows for more controllability compared to user-defined rules.

## Weaknesses

- The results presented in Table 1 shows that performance of CTG++ does not always perform better than baselines, and when it does, performance is only marginally better. is only marginally better.
- Evaluation is performed only on 1 dataset. How does this method compare when evaluated more widely?
- Exploiting LLMs is a core contribution of this work; in order to get a deeper understanding of the approach, it would be useful to investigate the effect of a weaker/stringer LLM. Would this lead to better results?
- The page formatting has been abused in order to try and fit into the 8 page limit which made it difficult to parse captions.

**Quality Of The Limitations Section:**

Additional details required

**Questions For Rebuttal:**

- How does this method compare when evaluated more widely?
- How often is the output from the LLM incorrect? Does the method have a means for detecting this?

**Robotics Focus:**

Highly relevant to robotics but no hardware experiments

**Summary Of Paper:**

This paper proposes CTG++, a new method to generate trajectories for traffic simulation. It presents the following novel elements:

- Scene-level conditional diffusion model, with multi-agent trajectories as output
- Guided sampling of trajectories from natural language

CTG++ builds upon Controllable Traffic Generation (CTG), presenting an architecture that models multi-agent trajectories using a spatio-temporal diffusion model. After denoising, it subsequently produces a set of trajectories for each agent. The paper leverages prompts from GPT4 to guide trajectory sampling. To overcome the lack of text-to-traffic data, they devised a method to inject language by prompting differentiable loss functions for different scenarios.

**Summary Of Recommendation:**

The approach is interesting, but the experimental results are underwhelming vs baselines. Currently, the results do not support the usefulness of the presented method.

---

### Official Review · Reviewer_QJon · 2023-07-20

**Confidence:** 4
**Originality:** Very Good
**Technical Quality:** Very Good
**Clarity Of Presentation:** Very Good
**Impact:** 4

**Recommendation:**

Strong Accept: I recommend accepting the paper and will argue for my recommendation even if other reviewers hold a different opinion.

**Review:**

I enjoyed reading this paper and thought the paper was well written. Each of the proposed improvement are natural, well motivated, and clearly described

The paper demonstrates strong emperical results in Table 1 and outperforms relevant baselines.

It Figure clearly illustrates both the generated programs and the effect of each design choice in the paper.

This paper tackles a timely problem that is of interest to many people at CoRL.

**Quality Of The Limitations Section:**

Limitations are addressed clearly

**Questions For Rebuttal:**

It might be interesting to show some more examples of synthesized programs -- to what extent does the prompted example program need to roughly reflect the cost function we want to use for guidance?

It might be interesting to try to the ULA sampler from [1] to combine the guidance function with diffusion sampling. Directly adding gradients during sampling has some theoretical issues that are addressed by running additional MCMC steps to incoperate the constraint.

[1] Reduce, Reuse, Recycle: Compositional Generation with Energy-Based Diffusion Models and MCMC

**Robotics Focus:**

Highly relevant to robotics but no hardware experiments

**Summary Of Paper:**

This paper proposes to use language to generate different traffic configurations. The authors use a LLM to generate a guidance function that is used to steer a diffusion model for traffic synthesis. The authors further propose architectures that enable effective learning of the diffusion model such as a agent-centric coordinate system and a series of agent pairwise and temporal attention blocks. The authors show the efficacy of the approach across many different tasks.

**Summary Of Recommendation:**

I enjoyed reading this paper and think it should be accepted.

---

### Author Response · Authors · 2023-08-13
**Rebuttal Summary**

We want to thank all the reviewers for their insightful comments and their positive evaluation of our work! We are delighted to see that the reviewers recognize that our paper is well-written (QJon, qk9Q), tackles a timely problem that is of interest to many people at CoRL (QJon), presents a novel approach (qk9Q, J6rY), and shows strong empirical results (QJon).

In response to the valuable feedback provided by the reviewers, we have undertaken several enhancements to our manuscript (see the uploaded revised version where we highlight the newly added text/sections in red). In particular, we've provided significantly more details on the language guidance component by offering comprehensive queries for five generated success example programs (Appendix F.3) and two generated failure example programs (Appendix F.4). Besides, we've run three repetitions for the quantitative experiments (with the same settings as in Section 4.3) comparing CTG++ against the strongest baseline CTG with standard deviations shown for our results, underscoring the significant advantages of CTG++ (refer to Appendix G - Table 3). Furthermore, we have enriched the discussion on the limitations and future work of our method (see the updated Section 5). This includes the absence of automatic GPT4 output error detection/repairing and the relatively slow scenario generation process.

Below, we address each reviewer's questions and concerns in detail. We are optimistic that our work will serve as a catalyst for the community, spurring advancements in language controllable and realistic traffic generation and, consequently, the evolution of autonomous driving development.

---

### Decision · Program_Chairs · 2023-08-30

**Decision:**

Accept (Oral)

**Comment:**

The paper proposes CTG++, a natural language-conditioned diffusion model for generate traffic configurations. CTG++ uses an LLM to generate a guidance function that then steers a diffusion model for traffic synthesis. The paper additionally proposes architectures that enable effective learning of the diffusion model such as an agent-centric coordinate system and a series of agent pairwise and temporal attention blocks. Experiments across different tasks show the efficacy of the approach.

Reviewers recognize that the paper is well-written (QJon, qk9Q), presents a novel approach (qk9Q, J6rY) that tackles a timely problem that is of interest to many people at CoRL (QJon). Post-rebuttal (on request by the reviewers), the paper has also been improved to include more examples and analysis of synthesized programs (QJon) and additional experiment details and discussions (qk9Q, J6rY, eWu2). Several reviewers have acknowledged these changes and upgraded their scores. There are still some concerns on that the evaluations could be made stronger (in particular, it's relevance beyond language-to-traffic, qk9Q, eWu2) – nevertheless, within scope of the applications presented, reviewer sentiment is overall positive, leaning "strong accept" with extra points to novelty.

I agree with reviewers that the ideas presented in this work (while predominantly evaluated in the context of traffic simulation) are novel and well-positioned to inspire others in the community to extend these ideas across other pertinent areas in robotics.